# Structural dynamics of melting and glass formation in a two-dimensional hybrid perovskite

Chumei Ye [1,2], Lauren N. McHugh [3] ✉, Pierre Florian [4], Ruohan Yu[5], Celia Castillo-Blas [1], Celia Chen[1,2], Arad Lang[1], Yuhang Dai [5,6], Jingwei Hou [7,8], David A. Keen [9], Siân E. Dutton [2] ✉ & Thomas D. Bennett [1,10] ✉

Hybrid organic-inorganic perovskites (HOIPs) have garnered significant attention for their crystalline properties, yet recent findings reveal that they can also form liquid and glassy phases, offering an alternative platform for understanding non-crystalline materials. In this study, we present a detailed investigation into the structural dynamics of the melting and glass formation process of a two-dimensional (2D) HOIP, $(S-(-)-1-(1-naphthyl)$ ethylammonium)$_2$PbBr$_4$. Compared to its crystalline counterpart, the glass exhibits superior mechanical properties, including higher Young's modulus and hardness. Our structural studies reveal that the liquid and glass formed from the 2D HOIP exhibit network-forming behaviour, featuring limited short-range order within individual octahedra, partial retention of metal-halide-metal connectivity between neighbouring octahedra, and residual structural correlations mediated by organic cations. We then combine in situ variable-temperature X-ray total scattering experiments, terahertz far-infrared absorption spectroscopy and solid-state nuclear magnetic resonance techniques to study the melting mechanism and the nature of the HOIP liquid obtained. Our results deepen the understanding of the structural evolution and property relationships in HOIP glasses, providing a foundation for their potential applications in advanced phase-change material technologies.

Hybrid organic-inorganic perovskites (HOIPs) are a class of three-dimensional (3D) materials with the general stoichiometry formula ABX$_3$, where A is a monovalent organic cation, B is a divalent metal ion and X is a monovalent halide or pseudohalide anion[1]. Their crystal structure consists of $[BX_6]^{4-}$ octahedra that are three-dimensionally connected through the corner X-site anions, with A-site cations occupying empty cuboctahedral cavities surrounded by eight corner-sharing octahedra. Beyond the archetypal 3D structure, oversized

[1]Department of Materials Science and Metallurgy, University of Cambridge, Cambridge CB3 0FS, UK. [2]Cavendish Laboratory, University of Cambridge, Cambridge CB3 0HE, UK. [3]Department of Chemistry, University of Liverpool, Liverpool L69 7ZD, UK. [4]CNRS, CEMHTI UPR3079 University of Orléans, Orléans 45000, France. [5]State Key Laboratory of Advanced Technology for Materials Synthesis and Processing, Wuhan University of Technology, Wuhan 430070, PR China. [6]Department of Engineering Science, University of Oxford, Oxford OX1 3PJ, UK. [7]School of Chemical Engineering, The University of Queensland, St Lucia, QLD 4072, Australia. [8]ARC Centre of Excellence for Green Electrochemical Transformation of Carbon Dioxide, Brisbane 4072, Australia. [9]ISIS Facility, Rutherford Appleton Laboratory, Harwell Campus, Didcot OX11 0QX, UK. [10]MacDiarmid Institute for Advanced Materials and Nanotechnology, School of Physical and Chemical Sciences, University of Canterbury, Christchurch 8140, New Zealand. ✉e-mail: L.N.Mchugh@liverpool.ac.uk; sed33@cam.ac.uk; tdb35@cam.ac.uk; thomas.bennett@canterbury.ac.nz

A-site organic cations typically form lower-dimensional phases such as two-dimensional (2D) Ruddlesden−Popper perovskites ($A_2BX_4$), in which organic bilayers and inorganic $[BX_6]^{4-}$ slabs are alternately stacked to form a layered structure[2]. 2D HOIPs have recently been in the spotlight mainly due to their superior long-term stability compared with their 3D counterparts[3]. The diverse and abundant selection of available organic molecules and the ionic compositions affords 2D HOIPs a wide compositional variation and tuneability towards a broad range of applications such as light-emitting diodes[4,5], photodetectors[6,7] and laser devices[8,9].

Glasses can be formed via a number of routes, the most common of which involves quenching a liquid from above its melting temperature ($T_m$) to below its glass transition temperature ($T_g$), thereby avoiding long-range structural rearrangement[10–12]. Hybrid glasses, the non-crystalline counterparts of typically crystalline hybrid materials assembled from inorganic and organic building units, are a burgeoning class of glassy materials that are distinct from the well-established inorganic, organic and metallic forms[13,14]. Initially developed from metal-organic frameworks (MOFs)[15] and coordination polymers (CPs)[16], these glasses possess similar metal-ligand bonding motifs to their crystalline cousins but lack the long-range order associated with crystallinity[17,18]. They thus benefit from the advantages of liquids and glasses such as transparency, isotropic properties and shaping abilities, broadening their chemistry and processing options, and have displayed great promise in areas such as gas separation[19–21] and ion transport[22,23]. Further hybrid glasses are thus highly desired, although the rare occurrence of stable MOF and CP liquids means that only a minimal number have been identified to date.

The long-held view that hybrid perovskites exclusively exist in a crystalline form has recently been challenged by the discovery of glass formation in HOIP materials[24]. An increasing number of HOIPs have been observed to melt at elevated temperatures and form glasses through quenching. Melting in HOIPs is a highly complex process influenced by their hybrid bonding nature, which includes ionic, covalent, hydrogen bonding, and various supramolecular interactions[25]. Unlike classical melting, which follows a straightforward equilibrium between solid and liquid states, melting in such hybrid materials often proceeds via intermediate phases. For instance, zeolitic imidazolate frameworks undergo a multi-step melting process, starting with a structural collapse into an amorphous phase, followed by recrystallisation into a denser structure, and ultimately transitioning to the liquid state[26]. Similarly, the diverse phase transitions observed in HOIPs, such as chain-melting in 2D systems, may further complicate their thermal behaviour by generating partially disordered intermediate phases[27]. These complexities highlight the importance for in-depth studies to elucidate and control the melting process of hybrid perovskites.

To date, examples of glass-forming HOIPs include 3D [TPrA] [M(dca)$_3$] (TPrA = tetrapropylammonium, M = $Mn^{2+}$, $Fe^{2+}$, $Co^{2+}$, dca = dicyanamide)[28], and several 2D systems such as chiral (S-/R-NEA)$_2$PbBr$_4$ (NEA = 1-(1-naphthyl)ethylammonium)[29], (1-MeHa)$_2$PbI$_4$ (1-MeHa = 1-methyl-hexylammonium)[30] and (MIPA)$_2$PbI$_4$ (MIPA = N-methyl iodopropylammonium)[31]. Among these, (S-NEA)$_2$PbBr$_4$ is one of the most extensively studied systems. Prior studies by Mitzi et al. have made progress in this area by investigating the reversible crystal-glass switching[29] and the kinetics of glass crystallisation[30]. Our earlier work also demonstrated that both chiral (S-/R-NEA)$_2$PbBr$_4$ and its non-melting racemic analogue can form glasses via ball-milling, offering an alternative to the conventional melt-quenching approach[32]. More recently, Singh et al. investigated the local structure of this HOIP in its molten and glassy phases, with a focus on metal-halide coordination and organic-inorganic interface interactions[33]. However, these studies have primarily focused on static structural properties, leaving the structural evolution during crystal-to-liquid-to-glass transitions largely unexplored.

Furthermore, research into the dynamics of melting and vitrification remains scarce, yet understanding the atomic-scale behaviours during these processes is essential. These insights not only enable the elucidation of the underlying structure-property relationships but also facilitate the unlocking of their full potential in advanced applications such as phase-change memory and computing[25].

In this article, we build upon previous studies to present a comprehensive investigation into the structural dynamics of (S-NEA)$_2$PbBr$_4$ during melting and glass formation. The microstructure of both the crystalline and glassy phases was examined using scanning transmission electron microscopy (STEM). The degree of structural disorder across various length scales in the glassy phase was further elucidated through X-ray total scattering and multidimensional magic angle spinning (MAS) solid-state nuclear magnetic resonance (ssNMR). To yield a complete picture of the melting mechanism, we employed a combination of in situ variable-temperature experiments, including pair distribution function (PDF) analysis, terahertz far-infra-red (THz/Far-IR) absorption spectroscopy, and multinuclear ssNMR. These complementary techniques provided detailed insights into the structural evolution and transition dynamics during the crystal-to-liquid-to-glass transformation. Finally, we extend our investigation to correlate the optical and mechanical properties of the glassy material with its structural features, thereby establishing new structure-property relationships. By integrating these analyses, this work sheds light on the microscopic mechanisms underpinning the glass formation in HOIPs and lays the foundation for the rational design of hybrid glasses with advanced functional properties.

## Results
### Crystal melting and glass formation of 2D HOIPs
Single crystals of (S-NEA)$_2$PbBr$_4$ were synthesised via a solvothermal reaction following previous literature procedures (see Methods section, Supplementary Figs. 1−2 and Supplementary Table 1)[34]. It crystallises in the monoclinic $P2_1$ space group, with inorganic $[PbBr_4]^{2-}$ layers of corner-sharing PbBr$_6$ octahedra separated by bilayers of chiral (S-NEA)$^+$ spacer cations (Fig. 1a). Thermogravimetric analysis (TGA) and differential scanning calorimetry (DSC) were carried out under an inert argon atmosphere to confirm the expected thermal behaviour (Supplementary Fig. 3a, b). The TGA trace of (S-NEA)$_2$PbBr$_4$ was featureless before thermal degradation, while the weight loss below 200 °C was less than 0.2%. The DSC measurement showed an endothermic response attributable to a solid-to-liquid transition, revealing a $T_m$ of ca. 176 °C and melting enthalpy ($\Delta H_m$) of ca. 51 J g$^{-1}$ (Fig. 1b). Liquids of (S-NEA)$_2$PbBr$_4$ were then quenched to yield a glassy phase $a_g$(S-NEA)$_2$PbBr$_4$ ($a_g$: melt-quenched glass). Ambient temperature powder X-ray diffraction (PXRD) confirmed the amorphous nature of $a_g$(S-NEA)$_2$PbBr$_4$, as only diffuse scattering was observed (Fig. 1c). The TGA trace of $a_g$(S-NEA)$_2$PbBr$_4$ had a two-stage mass loss after decomposition at ca. 210 °C (Supplementary Fig. 3c), consistent with the degradation of the crystalline analogue. The corresponding DSC heating experiment indicated a $T_g$ of ca. 68 °C (Fig. 1b), which is an intrinsic characteristic of glasses and corresponds to a reversible transition from a solid glassy phase to a viscoelastic supercooled liquid phase. Continued heating of the supercooled liquid above $T_g$ resulted in the crystallisation of (S-NEA)$_2$PbBr$_4$ at $T_x$ = 115 °C ($T_x$ is the onset temperature for the crystallisation peak) on the timescale of the DSC experiment, before reaching the liquid state again at higher temperature (Fig. 1b, Supplementary Figs. 4−5 and Supplementary Table 2). The dependence of $T_g$ on heating rate reveals a fragility index (m) of 33 for (S-NEA)$_2$PbBr$_4$ (Supplementary Fig. 6), which is intermediate between strong liquids such as silica (m = 20), and fragile ones, e.g., toluene (m = 105)[15] and triphenylphosphate (m = 160)[35]. The glass-forming ability of (S-NEA)$_2$PbBr$_4$ is also demonstrated by its $T_g/T_m$ ratio of 0.76, which exceeds the empirical Kauzmann 2/3 Law[36]. To further compare glass-forming tendencies, single crystals of chiral (R-

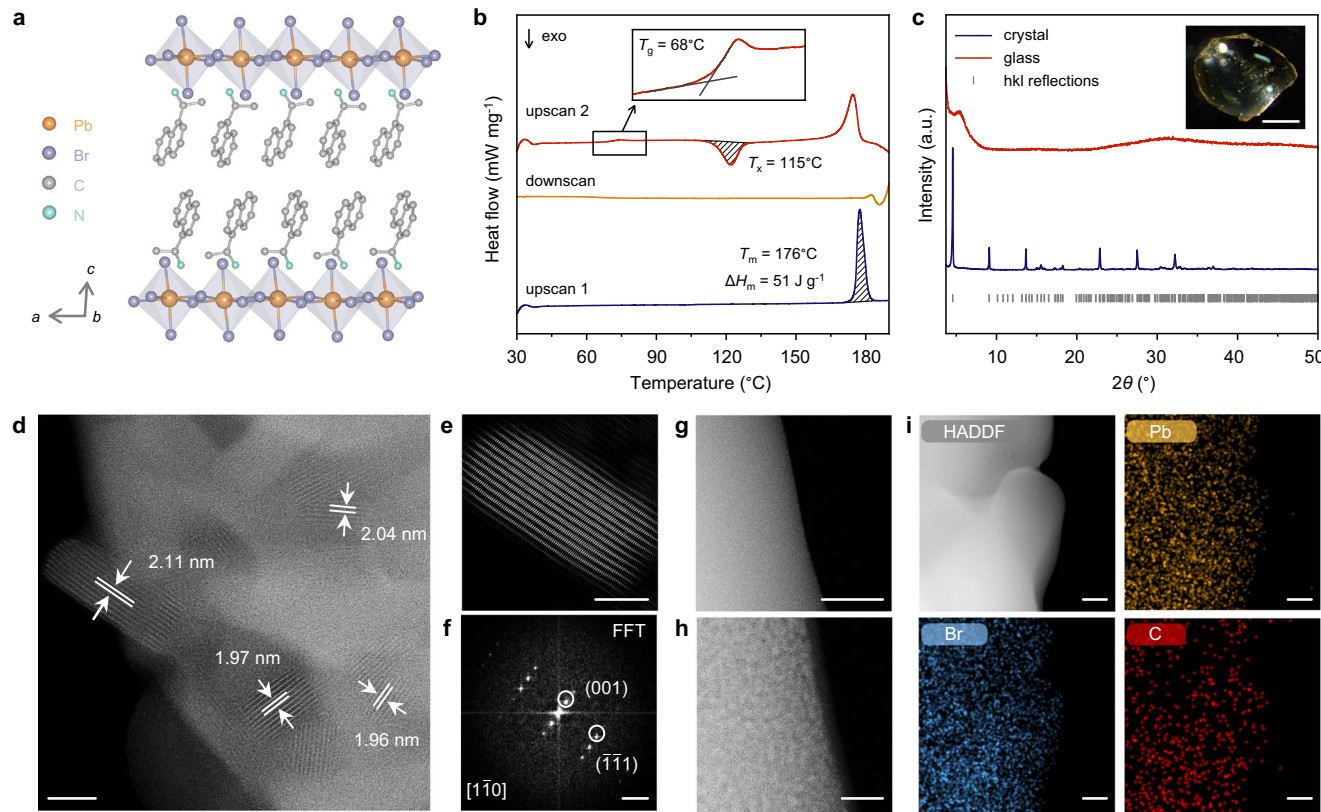

**Fig. 1 | Crystal-glass transformation and microstructural investigation.**
**a** Schematic diagram of the crystal structure of $(S\text{-NEA})_2PbBr_4$ viewed along the crystallographic $b$-axis. Atoms shown are Pb (orange), Br (purple), C (grey) and N (green). H atoms are omitted for clarity. **b** Differential scanning calorimetry (DSC) traces of $(S\text{-NEA})_2PbBr_4$ recorded under argon at a heating/cooling rate of 10 °C min$^{-1}$. The first heating upscan (navy) shows a melting endotherm at $T_m = 176$ °C, with an enthalpy of melting $\Delta H_m = 51$ J g$^{-1}$ (shaded). The second heating upscan (red) displays a glass transition at $T_g = 68$ °C and a crystallisation exotherm at $T_x = 115$ °C, with an enthalpy of crystallisation $\Delta H_x = -28$ J g$^{-1}$ (shaded). Inset: magnified view of the glass transition region in the second heating cycle. **c** Powder X-ray diffraction patterns of crystalline (navy) and glassy (red) $(S\text{-NEA})_2PbBr_4$, with symmetry-allowed Bragg reflections indicated by grey ticks. The intensity of the glass pattern is scaled by a factor of 10 to highlight the broad features. Inset: optical microscope image of the glass sample; scale bar, 1 mm. **d–f** Atomic-resolution imaging of crystalline $(S\text{-NEA})_2PbBr_4$ using scanning transmission electron microscopy (STEM): **d** High-angle annular dark-field (HAADF) STEM image showing the periodic lattice fringes with interplanar spacings of $ca.$ 2 nm; **e** Processed STEM image of an individual nanosheet oriented along the $[1\bar{1}0]$ zone axis, judged from (**f**) the corresponding fast Fourier transform (FFT) pattern. STEM imaging and energy-dispersive X-ray spectroscopy (EDS) elemental mapping of the glass, $a_g(S\text{-}$ $NEA)_2PbBr_4$: **g**, **h** HAADF-STEM images; **i** EDS elemental maps showing the distribution of Pb (yellow), Br (blue), and C (red). Scale bar, 20 nm (**d**); 10 nm (**e**, **h**); 1 nm$^{-1}$ (**f**); 50 nm (**g**, **i**). Source data are provided as a Source Data file.

$NEA)_2PbBr_4$ and racemic $(rac\text{-NEA})_2PbBr_4$ were also synthesised (Supplementary Figs. 7–8 and Supplementary Tables 3–4). $(R\text{-NEA})_2PbBr_4$ exhibited a similar $T_g/T_m$ value of approximately 0.76, whereas $(rac\text{-}$ $NEA)_2PbBr_4$ decomposed before reaching its melting point (Supplementary Figs. 9–10). This is attributed to differences in organic cation packing, hydrogen bonding, and associated lattice distortions between chiral and racemic HOIPs[29].

## Microstructural investigation

To investigate the microstructural evolution on glass formation, scanning electron microscopy (SEM) images of $(S\text{-NEA})_2PbBr_4$ and $a_g(S\text{-NEA})_2PbBr_4$ were obtained (Supplementary Fig. 11), coupled with energy dispersive spectroscopy (EDS) analysis (Supplementary Fig. 12). The Pb/Br molar ratio derived from SEM-EDS (Supplementary Table 5), along with the C, H, N element composition calculated from CHN microanalysis (Supplementary Table 6), reveals that both the inorganic and organic components are well preserved in $a_g(S\text{-NEA})_2PbBr_4$. Additionally, Fourier transform infra-red (FTIR) spectra of $(S\text{-}$ $NEA)_2PbBr_4$ crystal and glass confirm the integrity of organic cations retained in the glassy phase, though the hydrogen-bonding interactions with bromine atoms appear weakened (Supplementary Fig. 13)[30]. Double spherical aberration-corrected scanning transmission electron

microscopy (STEM) images of $(S\text{-NEA})_2PbBr_4$ and $a_g(S\text{-NEA})_2PbBr_4$ were collected to yield more insights into the atomic-scale structure. For $(S\text{-NEA})_2PbBr_4$, parallel line-like lattice planes were observed and showed a pattern periodicity of ca. 2 nm (Fig. 1d), corresponding to the interlayer spacing between inorganic octahedral sheets. Most of the layered domains have sizes in the several tens of nanometres and demonstrate preferential growth (Fig. 1e, f and Supplementary Figs. 14–15). Upon melting, the ordered structure of the crystal is lost, but the STEM images of $a_g(S\text{-NEA})_2PbBr_4$ present a nearly uniform bright-dark contrasting pattern at high magnification (Fig. 1g, h and Supplementary Fig. 16), which might imply some local structural ordering around Pb in the glassy phase. STEM-EDS was also used to map the elemental distribution throughout the glassy phase, demonstrating the absence of phase separation at the nanoscale level (Fig. 1i).

## Local structures of crystalline and glassy phases

To gain insights into the structural ordering over various length scales, ambient-temperature synchrotron X-ray total scattering was performed on both $(S\text{-NEA})_2PbBr_4$ and $a_g(S\text{-NEA})_2PbBr_4$ using the I15-1 beamline at the Diamond Light Source (Oxfordshire, UK) with a wavelength $\lambda$ of 0.161669 Å. As expected, the structure factor $S(Q)$ displayed intense Bragg peaks from the powdered crystal, including

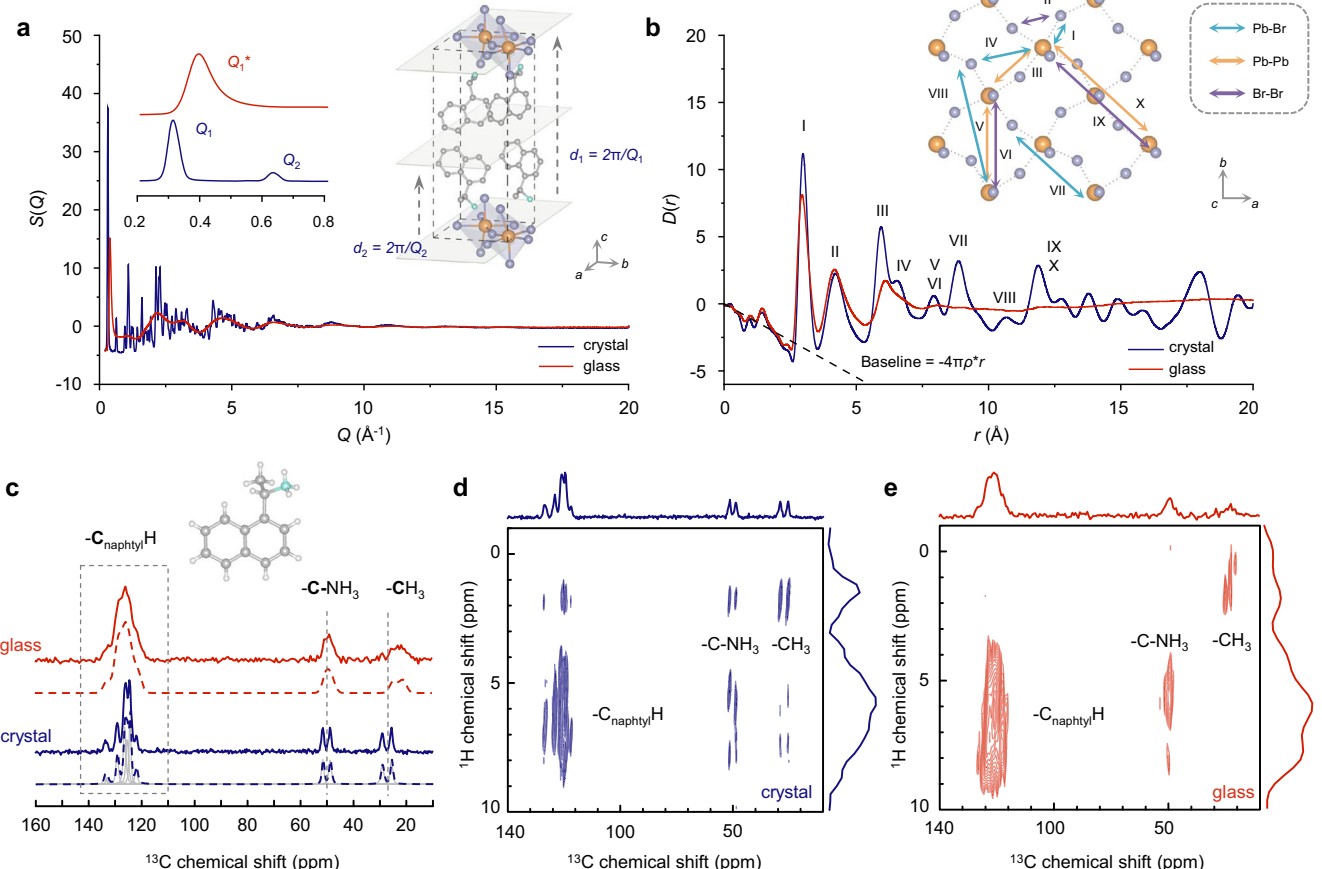

**Fig. 2 | Comparison of local structure between crystalline and glassy (S-NEA)$_2$PbBr$_4$. a** X-ray structure factors $S(Q)$ of crystalline and glassy (S-NEA)$_2$PbBr$_4$ obtained from total scattering experiments. Insets (left): magnified low-$Q$ region, highlighting the $Q_1$ and $Q_2$ features in the crystal, and the $Q_1^*$ feature in the glass. Inset (right): schematic illustration of interplanar spacings $d_1$ and $d_2$, corresponding to $Q_1$ and $Q_2$ features in the crystal structure, respectively. **b** Corresponding X-ray pair distribution functions $D(r)$. Peaks I-X are indexed according to interatomic correlations. Inset: schematic of peak assignments for Pb-Br (green), Pb-Pb (orange), and Br-Br (purple) distances. Contributions from atom pairs containing C, N, H are omitted due to their low X-ray form factors. **c** Experimental high-speed high-field MAS $^{13}$C NMR spectra (solid lines) collected at 17.6 T. Simulated spectra (dashed lines) and individual simulation components (grey lines) are vertically offset for clarity. Inset: chemical structure of the organic (S-NEA)$^+$ cation. Two-dimensional $^{13}$C{$^1$H} HETCOR spectra obtained with cross-polarization for (**d**) crystalline and (**e**) glassy (S-NEA)$_2$PbBr$_4$. In (**a–e**), data for crystalline (S-NEA)$_2$PbBr$_4$ and $a_g$(S-NEA)$_2$PbBr$_4$ are shown in navy and red, respectively. Source data are provided as a Source Data file.

two low-$Q$ peaks at $Q_1 = 0.32$ Å$^{-1}$ and $Q_2 = 0.64$ Å$^{-1}$ (Fig. 2a), which are caused by Bragg diffraction from the (001) and (002) lattice planes, respectively. Interpreting from a perspective of electron density modulation in real space, they describe density fluctuations with periods of 19.6 Å and 9.8 Å in the [001] lattice direction, and are thus structurally associated with the metal-organic-organic-metal ([BX$_6$] − A − A − [BX$_6$]) and the metal-organic ([BX$_6$] − A) building blocks within the layered A$_2$BX$_4$ perovskite, respectively. By comparison, only a single broader low-$Q$ peak was observed for $a_g$(S-NEA)$_2$PbBr$_4$ (Fig. 2a), which falls between $Q_1$ and $Q_2$ for the crystal and corresponds to a quasi-periodicity of 15.7 Å. This $Q_1^*$ peak may be indicative of electron density fluctuations between organic-metal-organic blocks, or it may be associated with weak layered ordering between inorganic components in the glass phase[33].

After data correction and Fourier transformation, the structure factor data were converted to the PDFs in the form of $D(r)$, conveying information about the characteristic atom-atom correlations in materials. Peaks in the PDF from the glass are mostly similar to those of the crystalline phase below 5 Å, although with a slightly decreased amplitude, indicative of local ordering on the sub-5 Å scale (Fig. 2b and Supplementary Fig. 17). Peak assignment was carried out using PDFgui, where the weighted partial PDFs for atom pairs, $g_{ij}(r)$, were calculated based on the published crystal structure (insets in Fig. 2b and

Supplementary Fig. 18)[34]. Interatomic correlations labelled I and II in Fig. 2b are attributed predominantly to the nearest Pb-Br and Br-Br atom pairs within an octahedron, respectively. Although the work by Singh et al.,[33] showed limited evidence of the change in local coordination, our results suggest a slight reduction in the average Pb-Br coordination number (CN$_{Pb-Br}$). Based on the integration of the correlation I in Fig. 2b, the CN$_{Pb-Br}$ is determined to be ca. 5.5 (Supplementary Fig. 17d, e and Supplementary Table 7), which might imply the presence of both six-fold and five-fold coordinated Pb atoms in $a_g$(S-NEA)$_2$PbBr$_4$. This is also accompanied by a slight decrease in the average Pb-Br bond length from 2.99 Å to 2.95 Å, as evidenced by the minor shift of the peak to lower-$r$ (Supplementary Fig. 17f). The peaks centred at 5.9 Å and 6.5 Å (labelled III and IV in Fig. 2b) in the crystalline-phase PDF can be primarily assigned to Pb-Pb and Pb-Br bonds between two neighbouring octahedra, respectively. However, in the PDF of the glassy phase, only an asymmetric broad peak was observed at 6.1 Å, which is reminiscent of the overlapping contributions from correlations III and IV in Fig. 2b. Despite the pronounced decrease in intensity relative to the crystal, it suggests the partial retention of metal-halide-metal (Pb-Br-Pb) linkage and corner-sharing connectivity in the glass.

The PDF of $a_g$(S-NEA)$_2$PbBr$_4$ exhibits peak broadening and substantially less prominent peaks above 7 Å as the crystal periodicity is

removed, confirming its complete transformation to the amorphous phase. In the intermediate-$r$ region (7 Å < $r$ < 15 Å), two weak peaks can be identified from the glassy-state PDF at 8 Å and 12 Å (Supplementary Fig. 17b), corresponding to Pb-Pb/Br-Br pair distances between two diagonal second-nearest-neighbour octahedra (labelled V and VI in Fig. 2b) and Pb-Pb/Br-Br pair distances between two linear second-nearest-neighbour octahedra (labelled IX and X in Fig. 2b), respectively. These features imply some connectivity between three neighbouring octahedra in the glassy phase that is reminiscent of the crystal structure. Further PDF analysis at high-$r$ reveals a damped oscillatory signal that persists to a distance of 80 Å for $a_g$(S-NEA)$_2$PbBr$_4$ (Supplementary Fig. 17c), which far exceeds the length scales of ionic liquids and molten salts as a result of charge neutrality[37]. This periodicity is comparable to those of network-forming glasses and liquids that stem from topological and chemical ordering[38,39]. These broad ripples have a period of approximately 16 Å over the extended length scale, which coincides with the scale of the first-sharp diffraction peak (FSDP, $Q_1$), namely the 15.7 Å oscillation for the glassy phase. The microscopic origin of the FSDP is a subject of debate and the interstitial void model proposed by S. R. Elliott has been considered as an elegant in-depth interpretation[40]. In this model, the FSDP is a chemical-order pre-peak resulting from the presence of interstitial volume, i.e., zones of low atomic occupancy, around cation-centred structural units[41–43]. For instance, the $Q_1$ positions for oxide and chalcogenide AX$_2$ glasses (A = Si, Ge; X = O, S, Se) generally span between 1 and 2 Å$^{-1}$ attributed to the cation-centred correlations, reflecting a quasi-periodicity of 3–7 Å in the glassy network. In contrast, the $Q_1^*$ oscillation of ca. 15.7 Å in the total scattering data for $a_g$(S-NEA)$_2$PbBr$_4$, is significantly broader, which is attributed to the large size of the A-site organic cation. In addition, to further explore how different preparation routes influence structural ordering, we compared the total scattering data of glasses formed via melt-quenching and ball-milling. The PDF data of the ball-milled glass, $a_m$(S-NEA)$_2$PbBr$_4$ ($a_m$ = mechanically amorphised), obtained from our prior study[32], reveal similar locally coordinated structural motifs to those in $a_g$(S-NEA)$_2$PbBr$_4$, with comparable Pb-Br coordination numbers (Supplementary Fig. 19). While these findings offer initial insights, further exploration of how different glass formation methods affect the overall structure and physical properties of HOIP glasses remains an intriguing avenue for future research.

To further investigate the behaviour of the organic cation, high-speed, high-field $^{13}$C and $^1$H MAS NMR spectra of (S-NEA)$_2$PbBr$_4$ crystal and glass were collected in tandem with first-principles density functional theory (DFT) calculations of the NMR parameters (Supplementary Figs. 20–21). The $^{13}$C spectra clearly show that multiple carbon environments are present in the crystalline phase: CH$_3$ groups appear around 27 ppm and CNH$_3$ around 50 ppm (Fig. 2c), both as two lines corresponding to the crystallographically (and magnetically) inequivalent molecules in the unit cell. The other carbon sites are grouped around 125 ppm, and only the resonance at 133 ppm may be attributed to the naphthyl carbon linked to the ethyl ammonium group, while all other resonances are too overlapping to be assigned to distinct carbon environments. The $^{13}$C NMR of $a_g$(S-NEA)$_2$PbBr$_4$ shows similar chemical environments, demonstrating that the structure of the organic cation is retained in the glassy phase after melt-quenching (Fig. 2c), in agreement with CHN microanalysis and FTIR results. The broadening of the lines in the $^{13}$C NMR corresponds to a slight distribution of carbon environments that are likely due to the increased intermolecular disorder in the glass.

Nevertheless, a clear difference observed in the glassy phase is the upfield shift of the CH$_3$ resonance in the $^{13}$C spectrum (Fig. 2c). This shift corresponds to the appearance of a second upfield, CH$_3$ line in the $^1$H spectrum (Supplementary Fig. 21b), which, in turn, exhibits correlations with neighbouring $^{13}$C signals in the two-dimensional $^{13}$C{$^1$H} heteronuclear correlation (HETCOR) spectra (Fig. 2d, e). This suggests

that the methyl group from the organic cation sees a different environment in the glass compared to the crystal, as can also be seen from the correlated $^1$H signals splitting into two resonances in the glass (Fig. 2e). $^1$H/$^1$H correlation (Supplementary Fig. 22) further confirms that this signal correlates with the resonances of the other protons (CH and NH$_3$) of the molecule, ruling out any phase separation and suggesting it likely arises from the molecular motion of -CH$_3$ groups. The appearance of a new $^1$H line at around 0 ppm is also clearly seen in both the $^1$H spectra and the $^1$H projections of the 2D HETCOR of the glassy phase (Supplementary Fig. 21). In the crystalline state, while the protons of the NH$_3$ group interact mostly with four bromine atoms, the protons of the CH$_3$ groups see two nearby ligands at 3.5 Å. The positional rearrangement of molecular cations with respect to each other in the glassy state could be the origin of this observation. Apart from the methyl group, it is also noticeable that the naphthyl CH groups at around 6 ppm in the $^1$H dimension in the crystalline form shift to higher frequencies, this could also point to increased H/H intermolecular interactions for those protons in the glassy phase.

## Structural dynamics in the HOIP liquid

To probe how the structure evolves during the crystal-liquid transformation, variable-temperature synchrotron X-ray total scattering data were collected while heating a crystalline sample of (S-NEA)$_2$PbBr$_4$ using the PETRA III beamline P02.1 in DESY (Hamburg, Germany) with a wavelength $\lambda$ of 0.20734 Å. Upon heating from 27 °C to 168 °C, both the position and intensity of the $Q_1$ and $Q_2$ features remained approximately constant, centred at 0.32 and 0.64 Å$^{-1}$, respectively (Supplementary Fig. 23). Further heating to 190 °C resulted in a pronounced intensity reduction, peak broadening and a shift in the $Q_1$-position to ca. 0.36 Å$^{-1}$, indicative of a similar contraction of quasi-periodicity as in the glassy phase. In the corresponding PDF data (Supplementary Fig. 24), as expected, the long-range ordering in the material, which persists below $T_m$, disappeared at 190 °C, confirming the complete transformation to the melt. The liquid-PDF bears a strong resemblance to the glassy-phase PDF, indicating that the structure of the liquid is well carried over into the glassy phase upon vitrification.

To quantitatively analyse how the metal-halide coordination geometry evolves upon heating, the integrated area and position of the intense Pb-Br PDF peak at ca. 3 Å were modelled as a function of temperature for (S-NEA)$_2$PbBr$_4$ (Fig. 3a, Supplementary Fig. 25 and Supplementary Table 8). The CN$_{Pb-Br}$ was almost unchanged below 168 °C and decreased to ca. 5.0 in the melt at 190 °C. Compared with other hybrid metal-organic liquids, this reduction in coordination number (17%) is reminiscent of ZIF liquids, where the average coordination number of Zn-N decreases from 4 to 3.6 (10%)[44], and liquids formed from 3D metal-bis(acetamide) frameworks, where the Co-Cl coordination decreases from 6 to 4.8 (20%)[37]. It is in contrast to the structural behaviours observed in 2D Zn(H$_2$PO$_4$)(HTr)$_2$ (HTr = 1,2,4-triazole)[45] and 1D zinc–phosphate–azole CPs[16], whose molten states contain discrete molecular fragments, i.e., without coordination bond preservation, referring to a more-ionic liquid-like structure. The molten (S-NEA)$_2$PbBr$_4$ also showed a decrease in the Pb-Br average distance compared to crystal, probably arising from enhanced increased electrostatic attractions between Pb centres and their nearest Br atoms (Fig. 3a and Supplementary Fig. 26).

To provide more insights into the melts at different temperature, high-temperature total scattering data were also collected at 180 °C and 190 °C using the I15-1 beamline at the Diamond Light Source (Oxfordshire, UK) and a $\lambda$ of 0.161669 Å, as used for the $a_g$(S-NEA)$_2$PbBr$_4$ (Supplementary Fig. 27-29). The structure factor data for the liquid at both temperatures are essentially similar, and both have a broad FSDP centred around $Q_1$ = 0.39 Å$^{-1}$. This corresponds to a quasi-periodicity of ca. 16.1 Å and approximates that of the glassy sample. A major difference between the liquids observed at 180 °C and 190 °C is the lower intensity of the FSDP at a higher temperature

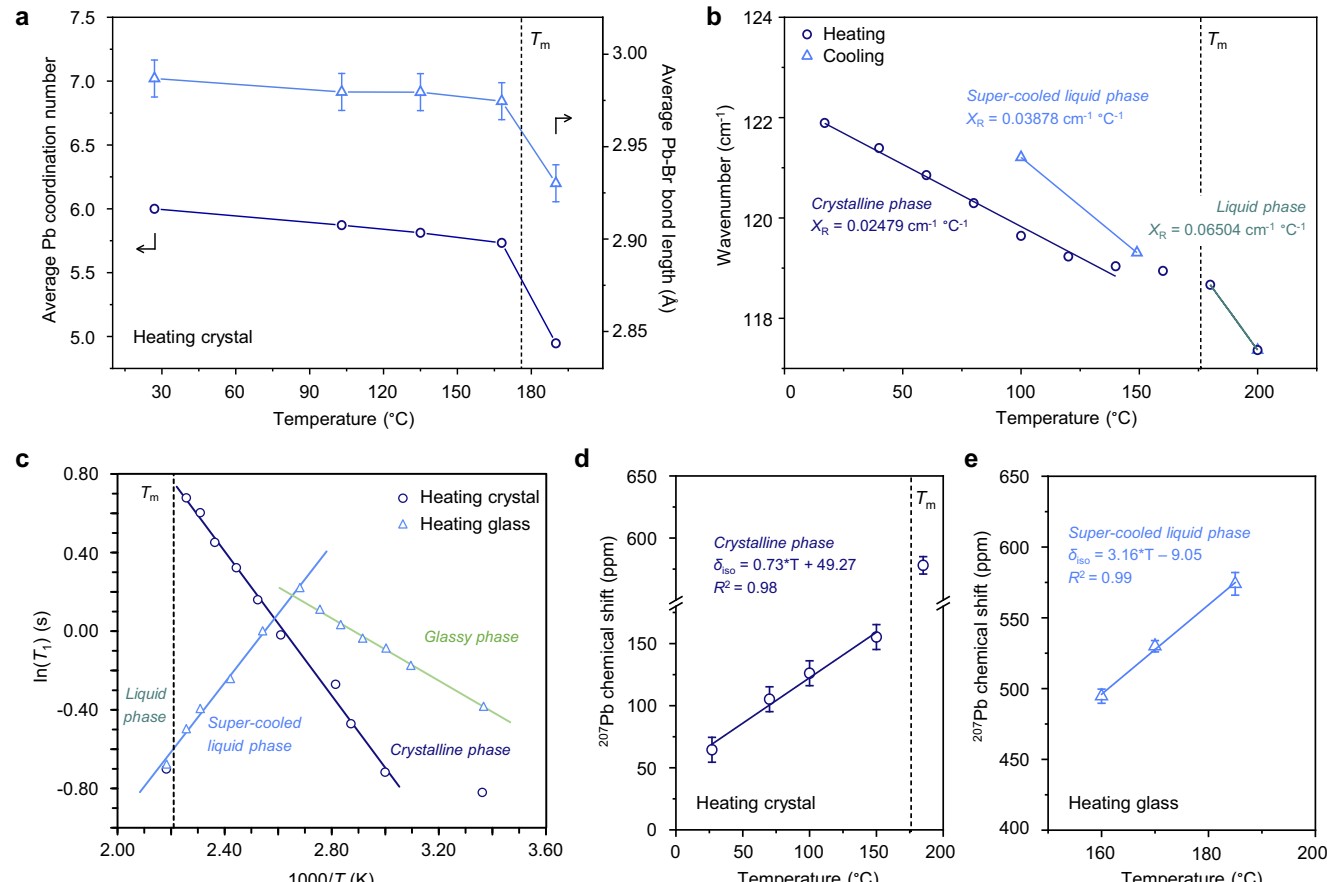

**Fig. 3 | Structure and dynamics in the liquid. a** Average Pb coordination number (navy circle, left axis) and average Pb-Br bond length (sky blue triangles, right axis) as a function of temperature, obtained from in situ variable-temperature X-ray total scattering measurements upon heating crystalline (*S*-NEA)₂PbBr₄. Solid lines connect the data points to guide the eye. **b** Temperature dependence of the Pb-Br stretching vibrational mode, measured by in situ THz/Far-IR spectroscopy during heating to 200 °C (navy circles) and subsequent cooling (sky blue triangles). Linear fits ($X_R$) indicate different phases: crystalline (navy line), liquid (teal line) and supercooled liquid (sky blue line). In situ variable-temperature NMR experiments during heating of (*S*-NEA)₂PbBr₄ crystal (navy circles) and glass (sky blue triangles): **c** plot of ¹H spin-lattice relaxation time ($T_1$) versus 1000/*T*, showing distinct linear regions assigned to the crystalline (navy), liquid (teal) and supercooled liquid (sky blue) phases; (**d, e**) ²⁰⁷Pb chemical shift ($\delta_{iso}$) as a function of temperature during heating of crystalline (**d**) and glassy (**e**) (*S*-NEA)₂PbBr₄, with solid lines representing linear fits for each phase. In (**a**) and (**d, e**), error bars represent fitting uncertainty in peak position determination. In (**a–d**), $T_m$ denotes the melting temperature of crystalline (*S*-NEA)₂PbBr₄ (176°C), as indicated by the black dashed line. Source data are provided as a Source Data file.

(Supplementary Fig. 27a), indicative of a less well-defined electron density variation through the structure. This is replicated by the smaller amplitude of the oscillatory signal observed in the high-*r* region of *D*(*r*) (Supplementary Fig. 28d), implying the degree of extended structural ordering in the melt decreases with increasing temperature. In the short- and intermediate-*r* regions, the PDFs of the liquid phases show minimal changes across different temperatures (Supplementary Fig. 28a–c), indicating that further heating has little effect on the local structural ordering within the melt. In comparison to the glassy phase, the average bond length for the shortest Pb-Br atom pair within an octahedron becomes shorter in the liquid state (Supplementary Fig. 28b). However, the distances for the Pb-Pb/Br-Br atom pairs between one octahedron and its second-nearest-neighbour octahedron become slightly longer (Supplementary Fig. 28c), indicating the corner-sharing connectivity becomes weaker in the high-temperature melts.

Temperature-resolved in situ THz/Far-IR vibrational spectroscopy was performed to study the bonding environment of (*S*-NEA)₂PbBr₄. All features remain largely unchanged in the region of 140–650 cm⁻¹ during heating and the subsequent cooling process (Supplementary Fig. 30). We assign the shifting mode at ~122 cm⁻¹ to the stretching vibration of Pb-Br within [PbBr₆]⁴⁻ octahedra, based on a previous

report[46]. On heating close to the melting temperature, the peak shape stays relatively constant, but a substantial redshift is observed for Pb-Br stretching. Linear curve fitting according to the Bose-Einstein statistic gives a slope ($X_R$) of 0.0248 cm⁻¹ °C (Fig. 3b), lower than that of Zn-N stretching in the crystalline phase of glass-forming ZIFs (~0.0255–0.0374 cm⁻¹ °C⁻¹)[47]. This may be associated with more rigid [PbBr₆]⁴⁻ coordination octahedra within HOIPs in contrast to the softer and more deformable ZnN₄ tetrahedra within ZIFs. Above $T_m$, the linear trend deviates from the trend of the crystalline solid, leading to an $X_R$ of 0.065 cm⁻¹ °C⁻¹. This is followed by a persistent blueshift of frequency upon cooling, resulting in an $X_R$ of 0.0388 cm⁻¹ °C⁻¹ for the supercooled liquid phase. This implies greater flexibility of the Pb environment in the liquid phase, in agreement with the partial dissociation of Br atoms from [PbBr₆]⁴⁻ octahedra concluded from PDF analysis.

In situ ¹H, ¹³C and ²⁰⁷Pb MAS NMR spectra upon heating crystalline samples up to 185 °C were collected to investigate the dynamic behaviour of the glass-former (*S*-NEA)₂PbBr₄ (Supplementary Figs. 31–33). In the ¹H spectra, no resolution is observed due to slow spinning, which does not average out ¹H/¹H dipolar interactions. Few changes are seen before 170 °C, after which intense sharpening of the signal indicates the presence of mobile protons on the 100 kHz timescale

(Supplementary Fig. 31). As a consequence, the $^1H/^{13}C$ dipolar couplings are averaged out by mobility as well and no cross-polarization could be obtained in this temperature range, however a solution-state sequence such as insensitive nuclei enhanced by polarization transfer (INEPT) recovers the $^{13}C$ signal (Supplementary Fig. 32). In the high-temperature melt, $^1H$ and $^{13}C$ spectra consist of very sharp lines ($^1H$ full width at half maximum of approx. 7 Hz) with positions matching fairly well with the ones obtained from DFT calculations when averaged over the various sites (Supplementary Fig. 34). This is a clear indication that the linker is intact in the melt and reorienting isotropically (random tumbling) at frequencies exceeding a few 100 kHz, effectively averaging out the $^1H/^1H$ and $^1H/^{13}C$ dipolar couplings, akin to the behaviour observed in solution-state conditions.

In situ $^1H$, $^{13}C$ and $^{207}Pb$ MAS NMR experiments were also performed on heating $a_g(S\text{-NEA})_2PbBr_4$ (Supplementary Figs. 35–37), which was obtained by quenching the molten sample inside the spectrometer after melting the crystal. In contrast to heating the parent crystalline sample, heating the glass leads to a more progressive evolution of the $^1H$ spectra as the line narrowing starts to be visible around 100 °C and continuously increases up to 185 °C (Supplementary Fig. 35). This different dynamical behaviour is also observed in the evolution of the $^1H$ spin-lattice relaxation times $T_1$ (Fig. 3c). On heating the crystalline form, $\ln(T_1)$ increases, linearly changing as a function of $1/T$, pointing to an Arrhenius behaviour with activation energy, $E_a = 15.3$ kJ mol$^{-1}$, before dropping down suddenly in the melt. When heating the glassy form, $\ln(T_1)$ increases up to approximately 105 °C and then decreases up to the melt, both changing linearly as a function of $1/T$ and indicating two types of motions with respective activation energies of 6.6 kJ mol$^{-1}$ and 14.6 kJ mol$^{-1}$. Positive slopes are the high temperature side of a fast movement with frequencies higher than the Larmor frequency (i.e. 200 MHz), whereas the negative slope corresponds to the "low temperature" side of a slow movement with frequencies below 200 MHz in the explored temperature range. "Solid-state" dynamics are, as expected, faster in the glassy form than in the crystalline one, and show no pre-melting phenomenon for the former. On the other hand, the super-cooled liquid phase above ca. 105 °C for the glassy form shows the onset of a low frequency movement which evolves continuously into the melt, as seen by both $^1H$ $T_1$ and line width evolutions. This onset temperature matches neither $T_g$ nor $T_c$, due to the difference in the time scales probed by DSC and NMR. $T_1$ and spectra obtained in the melt at 185 °C are the same upon heating the crystalline or the glassy form, showing that the thermal history is lost at this point, as expected for a high-temperature melt. It is also worth noting that the recrystallisation of the glassy sample upon heating was not detected in the in situ NMR experiments, which may be due to the continuous spinning impeding the nucleation and growth of crystalline grains.

Solid-state $^{207}Pb$ NMR spectroscopy is notoriously challenging due to the significant broadening typically associated with this nucleus. In our case, only the highly symmetric site within the crystal structure is detectable in the solid state, whilst the signal in the melt appears much more pronounced and substantially sharper relative to the solid. This reveals the high mobility of the Pb nuclei in the liquid phase and the rapid, random reorientation of Pb environment at frequencies far exceeding the linewidth (i.e. a few 100 kHz). The observed isotropic chemical shifts, $\delta_{iso}$, are close to the 361 ppm value reported for the methylammonium lead bromide perovskite $CH_3NH_3PbBr_3$[48]. Upon close inspection, linear trends between chemical shift and temperature are observed, with a slope of 0.73 ppm °C$^{-1}$ for the crystalline phase (Fig. 3d and Supplementary Fig. 33), close to the 0.905 ppm °C$^{-1}$ found for methylammonium lead chloride perovskites[49], and a higher slope of 3.16 ppm °C$^{-1}$ for the supercooled liquid phase (Fig. 3e and Supplementary Fig. 37). The variation in $\delta_{iso}$ for the crystalline state primarily arises from the intrinsic temperature sensitivity of the Pb

nucleus, caused by thermal expansion. The sudden increase in $\delta_{iso}$ (~400 ppm) upon melting, along with the stronger dependence observed in the liquid relative to the crystal, indicates a change in the chemical environment of Pb as it transitions from solid to melt and supercooled liquid. Assuming that the trends observed for lead oxides[50] apply here, this increase in chemical shift suggests a reduction in the average lead coordination number upon melting, in agreement with observations from in situ X-ray total scattering.

To establish a comprehensive understanding of the melting and glass formation process, Fig. 4 provides a visual summary of the key structural and dynamic changes in $(S\text{-NEA})_2PbBr_4$. In the crystalline phase, it adopts a layered perovskite structure, where $[PbBr_6]^{4-}$ octahedra form an extended corner-sharing network with sixfold coordination. Upon melting, long-range periodicity is lost, while short-range order persists. The Pb-Br coordination number decreases to an average of approximately 5.0, indicating slight decoordination in the melt, while Pb···Pb correlations between neighbouring octahedra remain detectable despite weakening, suggesting partial retention of corner-sharing connectivity. Structural disorder increases beyond three neighbouring octahedra, though extended correlations up to 80 Å in the PDF suggest the presence of organic-metal-organic units or remnants of layered order. Upon quenching, the glass retains the structural motifs of the melts, with a slight increase in Pb-Br coordination, reflecting a coexistence of five- and six-coordinated Pb environments. The $(S\text{-NEA})^+$ cations play a crucial role in stabilising the perovskite framework through hydrogen bonding. Upon melting, these interactions weaken as the organic cations gain mobility. Despite this, the distribution of cations remains confined within the framework, preventing complete structural collapse. In the glass, the cations are kinetically trapped in a disordered yet non-random configuration, with some interactions with the inorganic layers retained. The dynamic behaviour of both inorganic and organic components evolves significantly. In the crystal, the motion of both the $(S\text{-NEA})^+$ cations and the Pb-Br framework is relatively constrained. The Pb-Br stretching modes remain rigid at low temperatures but undergo a pronounced redshift approaching the melting point, indicating increased flexibility within the inorganic framework. In the liquid, high-temperature NMR reveals sharp $^1H$ and $^{13}C$ signals, suggesting rapid cation reorientation, while $^{207}Pb$ NMR shifts and peak narrowing confirm increased Pb mobility, consistent with THz/Far-IR and PDF analysis. Upon quenching, dynamic motion is arrested, though NMR reveals residual cation mobility, particularly in the methyl and ammonium groups, whereas the Pb-Br framework exhibits restricted atomic rearrangement, with partial retention of corner-sharing connectivity. The progressive loss of order, from the highly structured crystalline phase to the more fluid-like liquid state and ultimately to the kinetically trapped glass, underscores the complex nature of phase transitions in hybrid perovskites. Understanding these mechanisms not only advances our fundamental knowledge of hybrid perovskite glasses but also provides insight into tuning their properties for potential applications.

## Optical and mechanical properties

The facile and reversible crystal-glass transition of $(S\text{-NEA})_2PbBr_4$ makes it a potential candidate for applications such as phase-change memory, rewriteable data optical storage and non-volatile electronic memory, as data can be stored in the state of the material, i.e., crystalline (logic 1) or glassy (logic 0)[51]. Ambient-temperature ultraviolet-visible (UV-vis) spectra were recorded for both glassy and annealed crystalline films of $(S\text{-NEA})_2PbBr_4$ (Supplementary Fig. 38 and Supplementary Table 9). An absorption band with onset at 412 nm is present for the crystalline film, resulting in an experimental optical band gap ($E_g$) of 3.01 eV. In the glassy phase, the sample shifts towards a shorter wavelength, exhibiting an absorption onset of 365 nm and an $E_g$ of 3.44 eV, in accordance with reported values[29,32]. As suggested in an

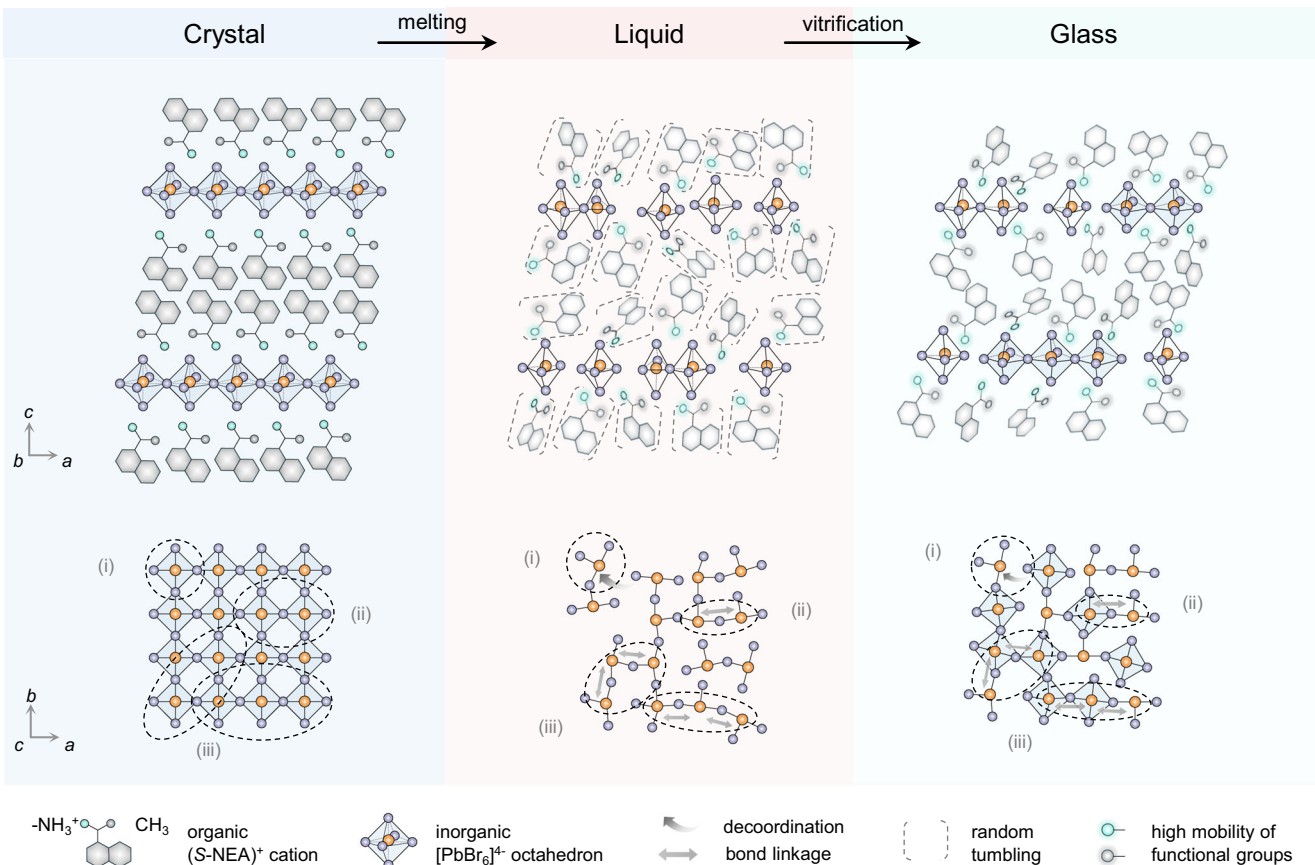

**Fig. 4 | Schematic illustration of the structural dynamics of melting and vitrification in the two-dimensional hybrid organic-inorganic perovskite ($S$-NEA)$_2$PbBr$_4$.** This figure depicts the transition from the crystalline to the liquid and glassy phases, highlighting key structural and dynamic changes. In the crystalline phase, [PbBr$_6$]$^{4-}$ octahedra form a well-ordered corner-sharing network, stabilised by hydrogen bonding with ($S$-NEA)$^+$ cations. Upon melting, long-range periodicity is lost, but short-range order persists with partial retention of Pb-Br coordination and connectivity between neighbouring octahedra. Increased molecular motion in the liquid phase leads to further structural disorder, though weak extended correlations remain. Vitrification kinetically traps the disordered liquid structure, resulting in a glass that retains key motifs of the melt. The labelled structural correlations indicate (i) within individual octahedra, (ii) between neighbouring octahedra, and (iii) between second-nearest-neighbour octahedra (either diagonal or linear).

earlier work by Jana et al.[34], the electronic band edge states of crystalline ($S$-NEA)$_2$PbBr$_4$ are governed by the inorganic framework, where the valence band maxima is dominated by Br states, and the conduction band minimum consists primarily of Pb states accompanied by spin-splitting as a result of inversion asymmetry. Based on the DFT-HSE0 + SOC calculations[34], the in-plane [100] crystallographic direction in the inorganic sublattice predominates the electronic band structure, whereas the out-of-plane [001] layer-stacking direction contributes little. While an intact [PbBr$_6$]$^{4-}$ octahedral network is present in the crystal, the average Pb-Br coordination number is calculated to be ca. 5.5 in the glassy phase, along with longer distances between second-nearest-neighbour octahedra. As such, we hypothesize that the undercoordinated Br atoms that dissociate from [PbBr$_6$]$^{4-}$ octahedra might be derived from the in-plane lead-bromine network, instead of those out-of-plane Br atoms.

Nanoindentation experiments were performed to study the mechanical properties of crystals of ($S$-NEA)$_2$PbBr$_4$ and bulk glasses of $a_g$($S$-NEA)$_2$PbBr$_4$ (Supplementary Fig. 39). Owing to the two-dimensional nature of the crystals, the reduced modulus ($E_r$) and hardness ($H$) were measured from the load-displacement data of indentations with the (001) lattice plane facing up. The $E_r$ and $H$ values of $a_g$($S$-NEA)$_2$PbBr$_4$ were significantly higher than those of its crystalline counterpart (Supplementary Table 10). To estimate the Young's modulus ($E$), a Poisson's ratio of 0.2 was assumed, resulting in $E = 2.30 \pm 0.10$ GPa for ($S$-NEA)$_2$PbBr$_4$ and $E = 5.29 \pm 0.04$ GPa for $a_g$($S$-

NEA)$_2$PbBr$_4$. These results provide preliminary insights into the mechanical properties of ($S$-NEA)$_2$PbBr$_4$; however, further studies are needed to fully understand features such as pop-ins, indentation size effects, and potential substrate influences in the experiments, presenting a compelling direction for future investigation. In line with other hybrid glasses, the $E$ and $H$ values of $a_g$($S$-NEA)$_2$PbBr$_4$ fall roughly midway between the upper bound of that expected for organic glasses and the lower bound for inorganic and metallic glasses (Supplementary Fig. 39e). Combined with its relatively low $T_g$, this glassy material demonstrates better processability than conventional inorganic glasses and MOF glasses.

## Discussion

The liquid and glass formed from the two-dimensional hybrid organic-inorganic perovskite exhibit network-forming behaviour, preserving structural correlations across multiple length scales. Despite the loss of long-range periodicity, short-range order remains in the non-crystalline phases, primarily through metal-halide local arrangements. This includes slight decoordination and subtle bond length contraction within individual octahedra. Between neighbouring octahedra, partial corner-sharing connectivity persists, as indicated by the retention of metal-halide-metal linkages spanning up to three octahedra. At longer distances, weak structural correlations extend up to 80 Å, with a quasi-periodicity of approximately 16 Å observed in the liquid and glassy phases. These extended correlations may originate

from the arrangement of organic-metal-organic motifs or residual layered ordering inherited from the crystalline phase.

The A-site organic cations remain intact in the liquid and glassy phase during melt-quenching and the disordered network structure of the liquid is carried over into the glassy phase upon vitrification. One major difference between liquids and glasses lies in the A-site organic cations: in the melt, the molecules undergo constant isotropic reorientation; in the glassy phase, the molecules undergo positional rearrangement compared to the crystal, accompanied by enhanced H/H intermolecular interactions involving the naphthyl group. The methyl and ammonium groups exhibit molecular motion with respect to the naphthyl rings, and their hydrogen-bonding interactions with bromine atoms are weakened. The dynamic behaviour of the organic cation in crystalline, glass and super-cooled liquid phases is examined by $^1H$ $T_1$ relaxation time analysis, revealing distinct activation energies for each phase. Finally, we correlate the structural features of different phases to their corresponding optical properties, which also suggests that the dissociation of in-plane halide atoms may be responsible for the decoordination of metal-halide octahedra.

Upon comparison with various glass-forming systems, the glass formed from hybrid perovskites displays a Young's modulus that is intermediate between organic, metallic glasses and inorganic glasses, and significantly higher than that of the parent crystalline perovskite. These results are a guide toward the fundamental understanding of the liquids and glasses formed from HOIPs and will support future efforts to take full advantage of their utility.

## Methods

### Materials

Lead(II) bromide (99.999%, Aldrich), (S)-(-)−1-(1-naphthyl)ethylamine (99%, Thermo Fisher), (R)-(+)−1-(1-naphthyl)ethylamine (99%, Merck) and 1-(1-naphthyl)ethylamine (99%, Fluorochem), hydrobromic acid (47 wt% HBr in $H_2O$, VMR Chemicals), diethyl ether (99.8 + %, Sigma-Aldrich) and methanol (99%, Fisher Scientific) were purchased as specified. All chemicals were used without further purification.

### Synthesis of hybrid perovskites

The synthesis of chiral (S-NEA)$_2$PbBr$_2$ and (R-NEA)$_2$PbBr$_2$ was based on a previously reported method in the literature[34]. Specifically, PbBr$_2$(0.48 mmol, 180 mg), and (S)-(-)- or (R)-(+)−1-(1-naphthyl)ethylamine (0.96 mmol, 156 μL) were dissolved in a mixture of hydrobromic acid (2.0 mL) and deionized water (4.8 mL) in a sealed vial and heated at 95 °C until fully dissolved. The resultant solution was gradually cooled to room temperature to obtain colourless crystals with flake-like morphology. Crystals of racemic (rac-NEA)$_2$PbBr$_4$ were obtained in a similar way by cooling a solution containing PbBr$_2$ (0.48 mmol, 180 mg), racemic 1-(1-naphthyl)ethylamine (0.96 mmol, 156 μL), hydrobromic acid (2.0 mL) and methanol (4.8 mL) from 95 °C to room temperature. The as-obtained crystals were then filtered, washed three times with diethyl ether, and vacuum-dried at 90 °C for 12 h.

### Preparation of (S-/R-NEA)$_2$PbBr$_4$ glasses

Unless otherwise stated, all glassy samples characterised in the study were prepared using the same method adapted from the literature[29]. The dried crystals were ground into fine powders using a mortar and pestle and placed onto a clean glass substrate. The substrate, along with the powders, was placed on a preheated hot plate at 190 °C for approximately two minutes, until the powders visibly transformed into a liquid state. The glass substrate was then removed from the hot plate using tweezers, quickly covered with another clean, room-temperature glass slide, and placed on a room-temperature metallic bench. Once fully cooled, the glassy sample was scraped from the glass substrate surface using a spatula. These procedures were carried out in a fume hood.

### Powder X-ray diffraction

Data were collected on a Bruker D8 ADVANCE diffractometer using Cu Kα radiation ($\lambda = 1.5418$ Å) and a LynxEye EX position-sensitive detector in Bragg-Brentano para-focusing geometry. Finely ground samples were compacted into 5 mm flat discs on a Si low-background substrate. Measurements were carried out at room temperature over the $2\theta$ range of 3°–70° for all samples, with a step size of 0.02° and measurement time of 0.750 s per step. Pawley refinements[52] were performed using TOPAS-Academic Version 7[53], and the lattice parameters were refined over the $2\theta$ range of 3°–70° against the values obtained from the published Crystallographic Information Files[34].

### Thermal characterisation

Differential scanning calorimetry experiments were conducted using a NETZSCH DSC 214 Polyma. Approximately 5–10 mg of ground samples were placed in an aluminium crucible and sealed using a hand press kit with a pierced lid. Before each sample scan, an empty aluminium crucible was used for correction. All samples were heated to 190 °C under an argon atmosphere to determine the melting point, then cooled to room temperature and reheated to 190 °C to determine the glass transition temperature. Heating and cooling rates of 10 °C min$^{-1}$ were employed unless otherwise stated. Data were processed using the Proteus Analysis software.

Simultaneous DSC-TGA measurements were performed on a TA Instruments SDT-Q65. Approximately 2–5 mg of ground samples were placed in an aluminium crucible and heated to 800 °C under an argon atmosphere with a rate of 10 °C min$^{-1}$. Data were processed using the TA Instruments Universal Analysis software package.

### CHN microanalysis

CHN combustion experiments were performed on crystalline (S-NEA)$_2$PbBr$_4$ and $a_g$(S-NEA)$_2$PbBr$_4$ powders to determine the elemental composition including carbon, hydrogen and nitrogen. Data were collected using a CE440 Elemental Analyser (EAI Exeter Analytical Inc.), and two measurements were conducted for each sample. The instrument was operated with tolerances of ± 0.2% for C and ± 0.1% for H and N. Compared to the crystalline counterpart, a slight decrease in the concentration of carbon (ca. 1.53%) and nitrogen (ca. 0.16%) was observed in the glass state (Supplementary Table 6).

### Infra-red spectroscopy

Fourier transform infra-red spectra for crystalline (S-NEA)$_2$PbBr$_4$ and $a_g$(S-NEA)$_2$PbBr$_4$ powders were recorded in transmittance mode between 600 and 4000 cm$^{-1}$, using a Nicolet iS50 FT-IR spectrometer. The background was subtracted from all spectra prior to analysis.

### Scanning electron microscope and energy dispersive spectroscopy

The surface morphologies of the as-synthesised bulk crystals, ground crystalline powders, melt-quenched bulk glass, and ground glassy powders were investigated using a high-resolution scanning electron microscope (FEI Nova Nano SEM 450). All samples were prepared by dispersing them onto a conductive double-sided carbon tape that was adhesively attached to an SEM specimen stub. The powder samples were then coated with a thin platinum layer using an Emtech K575 sputter coater to reduce charging. All images were acquired at an accelerating voltage of 15 kV unless otherwise stated. The weight percentages of Pb and Br were evaluated for the as-synthesised bulk crystals and melt-quenched glass based on SEM-EDS spectra, which were recorded at an accelerating voltage of 15 kV.

### Scanning transmission electron microscopy

STEM imaging and the corresponding energy dispersive spectroscopy were performed using a double spherical aberration corrected transmission electron microscope (Titan Cubed Themis G2 300). HAADF

and BF STEM images were simultaneously acquired. Data were processed using Digital Micrograph software and CrystalMaker equipped with CrystalDiffract.

### X-ray total scattering and pair distribution function analysis

X-ray synchrotron total scattering data were collected at the I15-1 beamline at the Diamond Light Source, UK ($\lambda = 0.161669$ Å). Finely ground samples of crystalline $(S\text{-NEA})_2PbBr_4$ and $a_g(S\text{-NEA})_2PbBr_4$ were loaded into borosilicate capillaries of 0.78 mm inner diameter to a height of approximately 3.6 cm, and sealed with plasticine. Ambient-temperature total scattering data were collected for the background (i.e. empty instrument), container (i.e. empty capillary) and all samples in a $Q$-range of 0.2–26 Å$^{-1}$. Various corrections for background, multiple scattering, container scattering, Compton scattering, and absorption were then performed in a $Q$-range of 0.2–20 Å$^{-1}$ to obtain structure factor $S(Q)$. The corrected total scattering data were subsequently Fourier transformed to obtain the real-space pair distribution function $G(r)$. In this work, the $D(r)$ form of pair distribution function was used to accentuate correlations at high-$r$ regions. All processing of the total scattering data was performed using the GudrunX programme following well-documented procedures[54–56]. High-temperature data were also collected upon heating the $(S\text{-NEA})_2PbBr_4$ melts at 180 °C and 190 °C using the I15-1 beamline. In addition, variable-temperature total scattering experiments were performed at the beamline P02.1 at PETRA III, DESY, Germany. Measurements were carried out on crystalline $(S\text{-}NEA)_2PbBr_4$ powders using an identical setup, though the borosilicate capillaries were sealed with super glue. Data were collected upon heating at 27, 103, 135, 168 and 190 °C and processed using equivalent measurements taken from an empty capillary heated to identical temperatures. Partial pair distribution functions for each atom pair were calculated based on the reported structure of $(S\text{-}NEA)_2PbBr_4$ using PDFgui.

### Terahertz far-infra-red absorption spectroscopy

Synchrotron THz/Far-IR spectra were collected at the THz/Far-IR beamline at the Australian Synchrotron, using a Bruker IFS 125/HR Fourier Transform spectrometer and a 6 µm thick multilayer Mylar beamsplitter. A liquid helium-cooled bolometer was used to obtain a high signal-to-noise ratio. All measurements were performed in attenuated total reflection (ATR) mode. Approximately 20 mg of crystalline $(S\text{-NEA})_2PbBr_4$ powders were mounted on the top of the diamond crystal, held in position by applied pressure and kept under flowing argon (ca. 20 mL min$^{-1}$). Temperature-resolved in situ spectra were recorded in 10 °C intervals from ambient temperature to 200 °C upon heating, and at 50 °C intervals down to 50 °C during cooling. Data were processed using extended ATR correction algorithms in the OPUS 8.0 software.

### Solid-state nuclear magnetic resonance spectroscopy

Samples of $a_g(S\text{-NEA})_2PbBr_4$ characterised during the in situ high-temperature experiments were obtained by quenching the sample inside the spectrometer at the end of the first run on the crystalline phase. An additional sample analysed at high-field high-speed was prepared by heating 50 mg of the crystalline form in a platinum crucible at 185 °C for 2 minutes and subsequently dipping it in liquid nitrogen.

All high-temperature magic-angle spinning solid-state NMR spectra of crystalline $(S\text{-NEA})_2PbBr_4$ and $a_g(S\text{-NEA})_2PbBr_4$ were collected on a Bruker Ascend wide-bore spectrometer operating at $^1$H (200.1 MHz), $^{13}$C (50.3 MHz), and $^{207}$Pb (41.9 MHz). Samples were packed in 4 mm diameter zirconia rotors, capped with zirconia caps, and spun at 10 kHz at all temperatures. $^1$H Hahn echo experiments were performed at an rf-field strength of 96 kHz, an interpulse delay of one rotor period (i.e. 100 µs), and a recycle delay between 1 s and

2.5 s, with 16 scans accumulated. $^1$H spin-lattice relaxation times $T_1$ were measured using a saturation-recovery sequence at an rf-field strength of 96 kHz and Hahn echo detection. $^{13}$C spectra in the solid state were obtained using a variable-amplitude cross-polarization (VACP) sequence with a 1.0 ms spin-lock time, using rf-field strengths of 30 kHz ($^{13}$C) and 20 kHz ($^1$H), and 1024 scans accumulated. $^{13}$C spectra of the liquid state were obtained using a J-based refocused INEPT sequence, with rf-fields of 77 kHz ($^{13}$C) and 96 kHz ($^1$H), 2Q excitation and reconversion times of 2.0 ms, and 256 transients accumulated. $^{207}$Pb spectra were recorded using a Hahn echo sequence at an rf-field strength of 80 kHz, an interpulse delay of 1 rotor period, and a recycle delay of 0.3 s. $^1$H and $^{13}$C spectra were referenced to TMS, and $^{207}$Pb spectra to a 0.1 M of Pb(NO$_3$) solution, with its peak set at 2941 ppm.

Additional $^1$H and $^{13}$C experiments were performed on a Bruker 17.6 T Neo spectrometer, operating at 750.0 MHz and 188.6 MHz, respectively, with spinning at 60 kHz and a controlled sample temperature of −20 °C. $^{13}$C{$^1$H} cross-polarization experiments were performed using a VACP 90–110% sequence, with a 0.35 ms spin lock time, rf-field strengths of 50 kHz ($^{13}$C) and 45 kHz ($^1$H), low-power $^1$H decoupling at 10 kHz, and a recycle delay of 1.5 s. A total of 1024 scans were accumulated for each $t_1$ increment over a 15 kHz indirect dimension, with 40 $t_1$ increments acquired using States mode. $^1$H DQ/SQ correlation experiments were performed using a DH$^3$ approach[57] with rf fields of 100 kHz, and excitation and mixing periods of 15 rotors cycles (250 µs).

First-principles NMR calculations with periodic boundary conditions were performed using the CASTEP code[58,59], which employs the planewave pseudopotential formalism of Kohn−Sham DFT. Electron correlation was modelled using the semi-local Perdew−Burke−Ernzerhof (PBE) exchange−correlation functional using DFT-TS van der Waals dispersion corrections[60]. The generalized gradient approximation (GGA)[61] was used along with the default "ultrasoft"[62] pseudopotentials of CASTEP 20.11 generated on-the-fly and a planewave cut-off energy of 570 eV. The Brillouin zone was sampled using a Monkhorst-Pack grid spacing of 0.07 Å$^{-1}$. Starting from the experimentally determined crystal structure, atomic positions were fully relaxed with convergence thresholds set to $5.0 \times 10^{-7}$ eV/atom for the total energy, $1 \times 10^{-2}$ eV Å$^{-1}$ for the maximum ionic force, 0.02 GPa for the maximum stress, and $5 \times 10^{-4}$ Å for the maximum ionic displacement. The NMR calculations were performed using the Gauge Including Projector Augmented Wave approach (GIPAW)[63,64] at the same cut-off energy of 570 eV. Calibration of $^{13}$C and $^1$H chemical shifts was obtained by performing calculations on amino acids according to the above-described conditions and comparing those to experimental reference values. This led to $\delta_{iso, \exp}(^{13}C) = -0.9589 \times \delta_{iso, cal}(^{13}C) + 165.541$ ($R^2 = 0.999$), and $\delta_{iso, \exp}(^1H) = -0.8483 \times \delta_{iso, cal}(^1H) + 26.534$ ($R^2 = 0.978$).

### Nanoindentation

Mechanical properties of the as-synthesised bulk crystals and melt-quenched bulk glasses were measured using an iNano Nanoindenter at ambient conditions. Layered crystal flakes, with the largest (001) plane facing up, were mounted on an aluminium stub using double-sided adhesive tape. Flat glass surfaces were obtained by pressing the liquids with a clean glass slide during melt-quenching, and mounted on a stub in the same manner. Prior to each measurement, at least four indents were performed on fused silica for calibration. Indentation experiments were conducted in the dynamic displacement-controlled mode, with a target depth of 2000 nm and a constant strain rate of 0.2 s$^{-1}$. Based on the Oliver-Pharr approach, the reduced modulus was calculated from the indentation curves (Supplementary Table 10). To determine the Young's modulus of the sample, the $E$ of the diamond indenter as 1141 GPa and its Poisson's ratio as 0.07, while assuming $v = 0.2$ for the sample, following previous studies on hybrid perovskites[28,65]. We also evaluated the sensitivity of $E$ to variations in $v$,

finding that increasing $v$ to 0.3 (as reported for lead iodide HOIPs[66]) resulted in only a 5% difference in the corresponding $E$ values. This confirms that the mechanical property estimates remain robust to reasonable variations in Poisson's ratio assumptions. After each sample test, the indenter tip was cleaned by performing at least four indents on a soft surface such as an old aluminium stub. The system was then recalibrated prior to the next sample test.

## UV-vis spectroscopy

UV-vis spectra of $a_g$(S-NEA)$_2$PbBr$_4$ and annealed crystalline (S-NEA)$_2$PbBr$_4$ films were recorded using a Cary 60 UV-vis spectrophotometer (Agilent). Glassy films were obtained by melt-quenching the crystalline powders sandwiched between two glass slides, while crystalline films were prepared by annealing the glassy films at 100 °C for 2 minutes. A background scan of two stacked glass slides was collected as a reference for each sample.

## Reporting summary

Further information on research design is available in the Nature Portfolio Reporting Summary linked to this article.

## Data availability

All data supporting the findings of this study are available within the article and its Supplementary Information. Source data are provided with this paper.

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

## Acknowledgements
The authors acknowledge Diamond Light Source, Rutherford Appleton Laboratory, United Kingdom, for the provision of synchrotron access to Beamline I15-1. We extend our gratitude to DESY (Hamburg, Germany), a member of the Helmholtz Association HGF, for the provision of experimental facilities. The authors would like to thank funding and support from St Edmund's College (C.Y.), the Royal Society for a university research fellowship, URF\R\211013 (T.D.B.), Leverhulme Trust for a Philip Leverhulme Prize (2019) (T.D.B. and L.N.M.), University of Liverpool (L.N.M.), the Winton Programme for the Physics of Sustainability (S.E.D.), the Cambridge Trusts and EPSRC Cambridge NanoDTC, EP/S022953/1 (C.C.), the Leverhulme Trust for a Research Project Grant, RPG-2020-005 (C.C.B. and T.D.B.), the Australian Research Council, DP230101901, LP220100309 and FT210100589 (J.H.), the Australian Government (J.H.), ARC Centre of Excellence for Green Electrochemical Transformation of Carbon Dioxide, CE230100017 (J.H.) and Blavatnik Family Foundation (A.L.). For the purpose of open access, the author has applied a Creative Commons Attribution (CC BY) license to any Author Accepted Manuscript version arising from this submission.

## Author contributions
T.D.B. and C.Y. conceptualised and designed the project. T.D.B., S.E.D., and L.N.M. supervised the project. T.D.B. acquired funding. C.Y. synthesised and characterised the samples. C.C.B. and C.Y. collected the total scattering data, and C.Y. analysed and interpreted the data with assistance from D.A.K. P.F. performed all NMR experiments and analysed the data together with C.Y. R.Y. and Y.D. collected the STEM data. J.H. collected the THz/Far-IR data, which was analysed by C.Y. with support from J.H. A.L. and C.C. contributed to useful discussions. C.Y. wrote the manuscript, and all authors contributed to the final version.

## Competing interests
The authors declare no competing interests.
