## [Peer Review file · Nature Communications]

Structural Dynamics of Melting and Glass Formation in a Two-dimensional Hybrid Perovskite

Corresponding Author: Professor Thomas Bennett

Version 0:

Reviewer comments:

Reviewer #1

(Remarks to the Author)

This paper addresses a study of the melting (and freezing) process in hybrid perovskites, using an impressive combination of experimental (in situ) techniques. More specifically, the authors consider the structural dynamics of melting in a 2D HIOp, which is thought to offer various advantages of the nowadays more common hybrid 3D perovskites. For both material classes, glass formation has very recently been demonstrated as a means to broaden the adaptability of functional properties as well as to enable alternative processing techniques. However, the range of practical materials available for glass formation is still very limited and mostly known only from phenomenology: a major issue towards understanding the utility of this prolific class of glasses as well as for their tailoring into real-world devices.

In depth studies as to why (or why not) and how melting occurs in such materials are therefore of great interest; they are the fundamental requisite for overcoming the above challenges. I therefore applaud the authors' work, in particular, the detail of observation leading to the scheme presented in Fig. 4.

I have some minor issues for the authors to discuss and consider in a potential revision of their manuscript:

the "melting" of (metastable?) high-entropy materials such as the current one, leading to a glass with higher density than the precursor crystal, is still a matter of exploration: for zeolitic materials, two-step processes were proposed in which the initial step is indeed not melting in the classical sense (involving a coexistence of two equilibrium phases), but a less-understood "collapse" reaction into a low-density amorphous state, followed by the actual melting process. Experimental evidence for this mechanism, however, remains scarce and is surely not within the scope of this paper, but perhaps the authors could briefly comment?

as a perspective, cyclic or modulated temperature scanning could help to differentiate reversible and irreversible reactions; for Fig. S5, it could help readers to show subsequent scans in a single panel for comparison; the caption says that different heating rates were applied – perhaps labels could be added in panel (c); the range of heating rates is not very broad, but the Arrhenius plot looks great.

Fig. S6 seems to correct a previous paper claiming an extraordinary stability parameter for a MOF-derived glass, what I greatly appreciate. Or is the red circle labelled in error? Given the above comment on collapse vs. melting, I remain hesitant about the stability parameters altogether and would recommend to not depict them in this manuscript. For example, if the collapse temperature of silicalites would be used to determine the stability parameter of vitreous silica (using the melting point of SiO₂ for reference), stability would also be close to unity.

sample thickness should be provided for UV-Vis spectra in Fig. S37, and eventually also for the photographs in Fig. S11
the nanoindentation experiments are interesting, but probably very preliminary ...? There are various pop-ins in the stress-strain curves; a statistical analysis could be of high interest to understand ductile yielding and stress localization in this material. Or are these experimental artefacts? Furthermore, there seems to be a notable indentation size effect, and perhaps a substrate artefacts effect, too, on the hardness data. But I agree that such in depth analyses would go beyond the current scope of the paper, aiming to provide only a first glimpse at the physical material properties.

there are some minor typos in the manuscript, such as on page 4 line 81. Please check.

Reviewer #2

(Remarks to the Author)

The paper describes the glass formation in 2D Hybrid Organic-Inorganic perovskite (S-(-)-1-(1-naphthyl)ethylammonium)2PbBr₄, a compound. In a previous report (Chem. Sci. 2024,15, 7198-7205) the same group has

provided insight into mechanically induced glass formation, here the authors have focused on the melt-quenched glass, melting mechanism and the dynamics of melt. The manuscript is well written and the authors were able to establish new structure-property relationships. This report also focuses on the fundamental understanding of crystal, glass and melt state using various techniques like STEM, multidimensional solid-state NMR In-situ variable-temperature experiments, including PDF analysis, terahertz far-infrared (THz/Far-IR) absorption. The work is interesting and provides new insights into the melting mechanism and glass formation hybrid glass-forming system. The work is suitable for publication in nature communications after revision as per the comments provided below:

1) Page 8, paragraph 2:

The coordination number values for melt and glass have been calculated from the PDF data. It is recommended that the fitting parameters and goodness of fit values should be provided in SI.

2) Page number 10, paragraph 3:

“a marked difference observed in the glassy phase is the appearance of a second, upfield, CH₃ line in the 1H spectrum (Fig. 2c) and correlates with neighboring 13C signals in the two-dimensional 13C{1H} heteronuclear correlation (HETCOR) spectra (Fig. 2d-e).”

Figure 2c is a 13C spectrum.

3) There are many typographical errors in the manuscripts (eg. “terahetrz” etc.)

4) Inconsistencies in the SI figure and caption, please check the SI thoroughly.

Examples:

Supplementary Information 2 -optical images, Figure 11

Supplementary Information 3 — Scanning electron microscopy images with EDS analysis, Figures 12 and 13.

Reviewer #3

(Remarks to the Author)

The authors conducted a thorough investigation of the structural dynamics during the melting and glass formation processes of the 2D HOIP (S-NEA)₂PbBr₄, which encompass short-range, some intermediate-range, and extended-range structural order. Their findings offer a deeper understanding of the structural evolution and interplay between structure and properties in HOIP glasses, thereby establishing a foundation for their future applications in advanced phase-change materials. I recommend an acceptance of this study after some revisions.

1. Ab initio molecular dynamics (AIMD) simulations are needed to add a different view and provide useful information of the microscopic structural evolution during the transitions, along with the experimental characterizations.
2. From a microscopic viewpoint, what insights can we get regarding structural reversibility after the glass-crystal phase transition with these detailed characterizations?
3. Fig 1b shows four phases in the second heating scan. One of them is between T_g and T_x, called viscoelastic supercooled liquid phase. Is it identical to the melting phase or can it find a unique place in Fig 4?
4. Define PDF when it first appears at line 81.

Reviewer #4

(Remarks to the Author)

In this paper the authors reported in-depth study of melt-quenched glasses of a HOIP, (S-NEA)₂PbBr₄, including: (1) DSC and TGA analysis that confirms the glass transition, and show the fragility; (2) the structural characterization of the melt-quenched glass by PDF, far-IR adsorption and NMR and (3) characterization of the dynamics associated with melted and glassy state. Overall the paper provides a thorough picture of the (S-NEA)₂PbBr₄ glass and should be published with minor revision. There are several points that needs clarification so that the audience would better understand the key message of this work.

1. The (S-NEA)₂PbBr₄ is a rather well-known material for its glass transition and as the authors have mentioned, there are several previous works on this, including one of the authors own research (ref 27). I think it would be better for the authors to discuss in more detail in the introduction section to clearly state what has been done for the (S-NEA)₂PbBr₄ glass and what information is newly provided in this paper. In this way the audience can clearly appreciate the novelty of this paper.
2. In ref 27, the authors reported that the (S-NEA)₂PbBr₄ can also give a glass by ball-milling, which has lower T_g. Could the author comments on the coordination number of the ball-milled glass compared to the melt-quenched glass? similarly, it would be good to show the comparison in short and medium range ordering
3. The quenching speed should also affect the medium range ordering of the Pd octahedral. Can the authors compare glasses made with different quenching rate?
4. In the mediate range regime, it seems that the 2D feature of the HOIP is preserved to a certain extent in the glassy state. If the same is melt-quenched under pressure (hot-press), would the resulting glass show anisotropic behavior?
5. The authors reported several characterizations, but the results are associated with several structural features (Pd octahedral distortion, etc). Could the author add one paragraph that discuss their findings in the order of structure features (i.e. firstly on Pd octahedral, then on Pd..Pd, then on organic portion, finally on the dynamics)? This will make the picture more clear.

Reviewer #5

(Remarks to the Author)

This is a thorough and interesting paper on the structural changes during melting and glass formation of a two-dimensional hybrid perovskite (S-NEA)₂PbBr₄. A wide range of analytical techniques are used for compositional and structural analyses, as well as selected thermal and mechanical characterization. The work provides detailed insights into the structural dynamics of the studied materials. While the synthesized material may not have any immediate applications and other hybrid perovskite glasses are already known, I think the fundamental understanding gained on the liquid and glass states presented here makes the work suitable for publication in Nature Communications after some revisions. In detail, I hope the authors can address the following points to strengthen their manuscript:

1. Some more discussion and clarification of the structural order of the studied liquid and glasses at different length scales is needed. In the Abstract, the authors refer to the materials having "limited short-range, some intermediate-range, and extended-range structural order". Glasses typically feature short-range order and by definition, no long-range (extended-range) order.
2. The novelty of the findings and the motivation for the chosen experimental design could be better highlighted at the end of the Introduction section.
3. Figure 1b shows the melting temperature of the original crystal in "upscan 1" to be at 176 C, while that of the crystal that forms upon re-heating the glass in "upscan 2" is at 115 C. Why is this melting temperature different? As I understand from the text, the same crystal structure as the original one forms upon crystallization of the glass. In any case, recrystallization of hybrid glasses remains fairly unusual, so I am wondering if the authors have tried to prepare glass-ceramics from these materials? (although I understand it may be out of scope for this work)
4. In Supplementary Table 5, what is the origin for the large difference in the elemental concentrations of "theoretical" vs. the measured ones, although the Br/Pb molar ratios are fine? Also, what is the meaning of the * symbol next to C and N in the table?
5. The differential correlation functions in Figure 2b shows that the intensity of peak II (Br-Br) is same in crystal and glass, but those of peaks I (Pb-Br) and III (Pb-Pb) are smaller in the glass. Do the authors have an explanation for this?
6. To improve clarity for readers (at least for me), I would have preferred to have the PDF data collected at the P02.1 beamline (p. 11) presented together with the other PDF data collected at the I15-1 beamline (pp. 7-8). It would be interesting to understand the liquid-PDF data before glass-PDF data. As a minor note, it is also not clear why high-temperature total scattering data had to be collected at two different beamlines.
7. The statement "temperature only has a small effect on the local ordering of melts" on p. 12 is not clear to me, when a 17% reduction in Pb-Br coordination number upon melting is reported on p. 11 based on the present data.
8. Why does the 207Pb MAS NMR signal appear to vanish at high temperature (Supplementary Figure 32), while it is stated in the text on p. 14 when discussing these data that "the signal in the melt appears much more pronounced and substantially sharper relative to the solid"? In comparison, the intensity of the 12C MAS NMR peak remains in the liquid state (Supplementary Figure S31). Also, the inset shown at top of Supplementary Figure 32 is not explained in the figure caption.
9. How sensitive are the reported mechanical properties to the assumption of a Poisson's ratio of 0.2? I would recommend reporting the reduced modulus instead of the Young's modulus to avoid relying on an assumed value without further evidence.
10. Figure 4 is a nice schematic presentation of the structural insights obtained on the various phases. However, the figure is not explained or discussed in much detail in the text. I think this would be important to do to assist readers in making the connections between the results of the different characterization methods.

Additional minor points:

11. Abbreviation PDF should be defined on p. 4.
12. "terahetrz" -> "terahertz" on p. 4
13. Supplementary Figures 23a, 25a, 26a: colors of the different lines (temperatures) are hard to distinguish.
14. Authors refer to a Supplementary Figure 39 on p. 16, but I could not find it. They seem to be referring to Supplementary Figure 38e?

Version 1:

Reviewer comments:

Reviewer #1

(Remarks to the Author)

My previous comments and suggestions were appropriately taken into account by the authors. Many thanks for this very nice

piece of work!

Reviewer #2

(Remarks to the Author)

Authors have revised the manuscript to my satisfaction. The additional results and critical structure property correlation has significantly improved the scientific rigour of the work present work. I recommend the publication of the revised manuscript.

Reviewer #3

(Remarks to the Author)

The manuscript has been carefully revised. I think it can be accepted in the present form.

Reviewer #4

(Remarks to the Author)

The authors have addressed my concerns and I support the publication of this paper.

Reviewer #5

(Remarks to the Author)

The authors have addressed all the comments satisfactorily. I recommend acceptance of this version for publication.

RESPONSE TO REVIEWERS' COMMENTS

Responses to Reviewer #1:

This paper addresses a study of the melting (and freezing) process in hybrid perovskites, using an impressive combination of experimental (in situ) techniques. More specifically, the authors consider the structural dynamics of melting in a 2D HIOF, which is thought to offer various advantages of the nowadays more common hybrid 3D perovskites. For both material classes, glass formation has very recently been demonstrated as a means to broaden the adaptability of functional properties as well as to enable alternative processing techniques. However, the range of practical materials available for glass formation is still very limited and mostly known only from phenomenology: a major issue towards understanding the utility of this prolific class of glasses as well as for their tailoring into real-world devices. In depth studies as to why (or why not) and how melting occurs in such materials are therefore of great interest; they are the fundamental requisite for overcoming the above challenges. I therefore applaud the authors' work, in particular, the detail of observation leading to the scheme presented in Fig. 4. I have some minor issues for the authors to discuss and consider in a potential revision of their manuscript.

We sincerely appreciate the reviewer's thoughtful feedback and their positive assessment of our work. We are delighted that the reviewer finds our study of the melting and glass formation processes in hybrid perovskites valuable, particularly the structural analysis and the proposed scheme in Figure 4. The reviewer's insightful comments have been instrumental in refining our manuscript, and we have carefully addressed each point in our detailed responses below. We hope that the revised manuscript meets the reviewer's expectations.

1. the "melting" of (metastable?) high-entropy materials such as the current one, leading to a glass with higher density than the precursor crystal, is still a matter of exploration: for zeolitic materials, two-step processes were proposed in which the initial step is indeed not melting in the classical sense (involving a coexistence of two equilibrium phases), but a less-understood "collapse" reaction into a low-density amorphous state, followed by the actual melting process. Experimental evidence for this mechanism, however, remains scarce and is surely not within the scope of this paper, but perhaps the authors could briefly comment?

We appreciate the reviewer's comment and suggestion. While $(S\text{-NEA})_2\text{PbBr}_4$ is not conventionally classified as a high-entropy material, we acknowledge the melting behaviour of such hybrid glass formers can involve multiple steps, such as structural collapse and phase transitions. These features may share some similarities with entropy-driven effects observed in high-entropy systems. Although a detailed exploration of this mechanism is beyond the scope of the current study, we have incorporated a discussion in the introduction to reflect the

complexity of the melting process in hybrid materials and its potential as a direction for future research.

- Section added to manuscript on Page 3 and Page 4:

An increasing number of HOIPs have been observed to melt at elevated temperatures and form glasses through melt-quenching. Melting in HOIPs is a highly complex process influenced by their hybrid bonding nature, which includes ionic, covalent, hydrogen bonding, and various supramolecular interactions²⁵. Unlike classical melting, which follows a straightforward equilibrium between solid and liquid states, melting in such hybrid materials often involves intermediate phases. For instance, zeolitic imidazolate frameworks (ZIFs) undergo a multi-step melting process, starting with a structural collapse into an amorphous phase, followed by recrystallisation into a denser structure, and ultimately transitioning to the liquid state²⁶. Similarly, the diverse phase transitions observed in HOIPs, such as chain-melting in 2D HOIPs, may further complicate their melting behaviour by producing partially disordered intermediate phases²⁷. These complexities highlight the need for in-depth studies to better understand and control the melting process in hybrid perovskites.

- Three references added to manuscript accordingly:

25. Singh, A. & Mitzi, D. B. Emergence of melt and glass states of halide perovskite semiconductors. *Nat. Rev. Mater.* in press (2025).

26. Bennett, T. D. et al. Hybrid glasses from strong and fragile metal-organic framework liquids. *Nat. Commun.* **6**, 8079 (2015).

27. Seo, J., Kim, H., Lee, S., Park, J., Choi, Y., Lee, K. et al. Colossal barocaloric effects with ultralow hysteresis in two-dimensional metal-halide perovskites. *Nat. Commun.* **13**, 2536 (2022).

2. as a perspective, cyclic or modulated temperature scanning could help to differentiate reversible and irreversible reactions; for Fig. S5, it could help readers to show subsequent scans in a single panel for comparison; the caption says that different heating rates were applied – perhaps labels could be added in panel (c); the range of heating rates is not very broad, but the Arrhenius plot looks great.

We appreciate the reviewer's comment and suggestion. Below, we address each point in detail.

(1) Cyclic temperature scanning: To illustrate the reversible nature of the crystal-glass transition, we have performed three heating-cooling cycles on crystalline (S-NEA)₂PbBr₄ using DSC. The corresponding data have been added as Supplementary Figure 3d in the Supplementary Information.

- Supplementary Figure 3d added to the Supplementary Information:

Supplementary Figure 3. Simultaneous DSC-TGA experiment on $(S-NEA)_2PbBr_4$. (a) crystalline $(S-NEA)_2PbBr_4$ in argon, showing a decomposition temperature of *ca.* 220°C and a melting temperature of *ca.* 180°C; (b) crystalline $(S-NEA)_2PbBr_4$ in air, showing a decomposition temperature of *ca.* 215°C and a melting temperature of *ca.* 178°C; (c) $\alpha_g(S-NEA)_2PbBr_4$ in argon, showing a decomposition temperature of *ca.* 210°C; (d) Cyclic DSC heating-cooling profiles of crystalline $(S-NEA)_2PbBr_4$ in argon, demonstrating the reversible crystal-glass transition. The inset shows the glass transition behaviours of the resultant glass after heating-cooling cycles.

(2) Subsequent scans in a single panel for Fig. S5: We have updated Supplementary Figure 6 to present the DSC scans more clearly. Supplementary Figures 6a-f now present the full DSC profiles of each sample used for the fragility measurement, including the first heating, first cooling, and second heating scans.

(3) Labels for different heating rates in panel (c): To enhance clarity, we have updated Supplementary Figure 6g by adding clear labels for the various heating rates applied during

the experiments.

(4) Range of heating rates: The heating rates in this study were selected based on the technical capabilities of our experimental DSC setup. Despite the limited range, it remains sufficient to derive meaningful insights. We are pleased that the reviewer found the Arrhenius plot informative.

- Previous Supplementary Figure 5 in the Supplementary Information:

Supplementary Figure 5. Fragility of the glass-forming liquid $(S-NEA)_2PbBr_4$. DSC profiles of the (a) first-heating, (b) first-cooling, and (c) second-heating scan of $(S-NEA)_2PbBr_4$ in argon. The first heating rates for all samples were $40\text{ }^{\circ}C\text{ min}^{-1}$, while various cooling/heating rates (q_h) ranging from $15\text{ }^{\circ}C\text{ min}^{-1}$ to $40\text{ }^{\circ}C\text{ min}^{-1}$ were used for different run. (d) Fragility of $(S-NEA)_2PbBr_4$ determined from the dependence of fictive temperature (T_f) on the heating rate (q_h) by using DSC.

- Revised Supplementary Figure 6 in the Supplementary Information:

Supplementary Figure 6. Fragility of the glass-forming liquid $(S\text{-NEA})_2\text{PbBr}_4$. (a-f) Full DSC profiles of the crystalline $(S\text{-NEA})_2\text{PbBr}_4$ samples in argon, showing first-heating, first-cooling, and second-heating scans. The first heating rates for all samples were 40 °C min⁻¹, while different first cooling and second heating rates (q_h) were applied for each sample, including 15 °C min⁻¹ for sample 1, 20 °C min⁻¹ for sample 2, 25 °C min⁻¹ for sample 3, 30 °C min⁻¹ for sample 4, 35 °C min⁻¹ for sample 5 and 40 °C min⁻¹ for sample 6. (continued)

(continued) (g) Comparison of the second heating DSC scans of samples at different heating rates. (h) Fragility (m) of glass-forming $(S-NEA)_2PbBr_4$ liquid, which is determined from the dependence of the fictive temperature (T_f) on the heating rate (q_h) using DSC.

(3) Fig. S6 seems to correct a previous paper claiming an extraordinary stability parameter for a MOF-derived glass, what I greatly appreciate. Or is the red circle labelled in error? Given the above comment on collapse vs. melting, I remain hesitant about the stability parameters altogether and would recommend to not depict them in this manuscript. For example, if the collapse temperature of silicalites would be used to determine the stability parameter of vitreous silica (using the melting point of SiO₂ for reference), stability would also be close to unity.

We thank the reviewer for their comment and suggestion. The labelled red circle in Supplementary Figure 6 corresponds to data extracted from the work by Bennett *et al.* (*J. Am. Chem. Soc.*, 2016, 138, 3484-3492), where the authors reported a melting temperature of 863 K and a glass transition temperature of 565 K for ZIF-4. In that study, heating ZIF-4 first led to structural collapse to an amorphous phase at ~600 K, followed by recrystallisation into a dense ZIF-zni framework prior to melting at 863 K. Given this, we acknowledge the reviewer's concerns regarding the use of stability parameters in systems where collapse, rather than classical melting, occurs. To avoid potential misinterpretation, we have removed the previous Supplementary Figure 6 from the Supplementary Information and revised the corresponding discussion in manuscript.

- Previous text in manuscript on Page 5:

The glass-forming ability of $a_g(S-NEA)_2PbBr_4$ is also evident from the higher ratio T_g/T_m (0.76) with respect to the empirical Kauzmann "2/3 Law"³³, and is similar to those reported for

zeolitic imidazolate framework (ZIF) glasses ($T_g/T_m = 0.65-0.85$)¹⁵ and $a_g[\text{TPrA}][\text{M}(\text{dca})_3]$ ($T_g/T_m = 0.79-0.93$)²⁵ (Supplementary Fig. 6).

- Revised text in manuscript on Page 6:

The glass-forming ability of $a_g(\text{S-NEA})_2\text{PbBr}_4$ is also demonstrated by its T_g/T_m ratio of 0.76, which exceeds the empirical Kauzmann “2/3 Law”³⁶.

(4) sample thickness should be provided for UV-Vis spectra in Fig. S37, and eventually also for the photographs in Fig. S11.

We appreciate the reviewer’s suggestion. To provide accurate information, we have measured the thickness of the film samples used in the UV-Vis experiment using a micrometer screw gauge. The results have been included in Supplementary Table 9 in the revised Supplementary Information. To ensure reliability, each film sample was measured at three different positions, and the average thickness was calculated.

- Supplementary Table 9 added to the Supplementary Information:

Supplementary Table 9. Thickness measurements of samples used in the UV-Vis experiment.

Sample	Thickness (mm)*			Average
	Position 1	Position 2	Position 3	
$(\text{S-NEA})_2\text{PbBr}_4$	0.05	0.04	0.06	0.05
$a_g(\text{S-NEA})_2\text{PbBr}_4$	0.04	0.05	0.07	0.05

* The thickness of each film sample was measured using a micrometer screw gauge with a precision of 0.01 mm, and the average value was calculated based on three measurements at different positions.

Additionally, we have provided the thickness of the film samples shown in the optical images in Supplementary Figure 1 (previous Supplementary Figure 11) for completeness.

- Previous caption of Supplementary Figure 11:

Supplementary Figure 11. Optical images of (a-b) as-synthesised $(\text{S-NEA})_2\text{PbBr}_4$ crystals; $a_g(\text{S-NEA})_2\text{PbBr}_4$ formed in (c) argon or (d) air; (e) $a_g(\text{S-NEA})_2\text{PbBr}_4$ film sandwiched between two glass slides and (f) the corresponding annealed $a_g(\text{S-NEA})_2\text{PbBr}_4$ film; (g-h) as-synthesised $(\text{R-NEA})_2\text{PbBr}_4$ crystals; (i) as-synthesised $(\text{rac-NEA})_2\text{PbBr}_4$ crystals.

- Revised caption of Supplementary Figure 1:

Supplementary Figure 1. Optical images of the as-synthesised, glassy and recrystallised samples. (a-b) As-synthesised $(S\text{-NEA})_2\text{PbBr}_4$ crystals. (c-d) $a_g(S\text{-NEA})_2\text{PbBr}_4$ formed in argon and air, respectively. (e) $a_g(S\text{-NEA})_2\text{PbBr}_4$ film sandwiched between two glass slides and (f) the corresponding recrystallised $a_g(S\text{-NEA})_2\text{PbBr}_4$ film. (g-h) As-synthesised $(R\text{-NEA})_2\text{PbBr}_4$ crystals; (i) As-synthesised $(rac\text{-NEA})_2\text{PbBr}_4$ crystals. In (e) and (f), the average thicknesses of the film samples were measured using a micrometer screw gauge with a precision of 0.01 mm, yielding approximately 0.09 mm and 0.11 mm, respectively.

(5) the nanoindentation experiments are interesting, but probably very preliminary ...? There are various pop-ins in the stress-strain curves; a statistical analysis could be of high interest to understand ductile yielding and stress localization in this material. Or are these experimental artefacts? Furthermore, there seems to be a notable indentation size effect, and perhaps a substrate artefacts effect, too, on the hardness data. But I agree that such in depth analyses would go beyond the current scope of the paper, aiming to provide only a first glimpse at the physical material properties.

We sincerely thank the reviewer for their insightful comment on the nanoindentation experiments. While our results provide an initial exploration of the mechanical properties of these materials, we acknowledge that a more detailed statistical analysis of stress-strain behaviour would provide valuable insights. Possible substrate-related artifacts could further influence the observed hardness data. As the reviewer noted, an in-depth investigation of these aspects is beyond the scope of the present work but represents an important direction for future research. To acknowledge these limitations, we have added a brief discussion in the revised manuscript.

- Section added to manuscript on Page 18:

These results provide preliminary insights into the mechanical properties of $(S\text{-NEA})_2\text{PbBr}_4$; however, further studies are needed to fully understand features such as pop-ins, indentation size effects, and potential substrate influences in the experiments, presenting a compelling direction for future investigation.

(6) there are some minor typos in the manuscript, such as on page 4 line 81. Please check.

We thank the reviewer for bringing this to our attention. We have carefully reviewed the manuscript and the Supplementary Information for typographical errors and corrected them where necessary, which are shown as follows.

In the manuscript:

- Page 2 line 18 and Page 4 line 81:

Before: terahetrz

After: terahertz

- Page 15 line 366:

Before: harness

After: hardness

In the Supplementary Information:

- Supplementary Figures 31, 33 and 36:

Before: ¹²C MAS NMR

After: ¹³C MAS NMR

Responses to Reviewer #2:

The paper describes the glass formation in 2D Hybrid Organic-Inorganic perovskite (S-(-)-1-(1-naphthyl)ethylammonium)₂PbBr₄, a compound. In a previous report (Chem. Sci. 2024,15, 7198-7205) the same group has provided insight into mechanically induced glass formation, here the authors have focused on the melt-quenched glass, melting mechanism and the dynamics of melt. The manuscript is well written and the authors were able to establish new structure-property relationships. This report also focuses on the fundamental understanding of crystal, glass and melt state using various techniques like STEM, multidimensional solid-state NMR In-situ variable-temperature experiments, including PDF analysis, terahertz far-infrared (THz/Far-IR) absorption. The work is interesting and provides new insights into the melting mechanism and glass formation hybrid glass-forming system. The work is suitable for publication in nature communications after revision as per the comments provided below:

We are grateful for the reviewer's recognition of our work and for acknowledging the significance of our findings. We appreciate the reviewer's positive remarks on the structure-property relationships established in this study and their endorsement of the manuscript's suitability for *Nature Communications*. The constructive feedback provided has been highly valuable in enhancing the clarity and impact of our work. Below, we offer point-by-point responses to each of the reviewer's suggestions and have made the necessary revisions accordingly.

1) Page 8, paragraph 2: The coordination number values for melt and glass have been calculated from the PDF data. It is recommended that the fitting parameters and goodness of fit values should be provided in SI.

We appreciate the reviewer's thoughtful suggestion. In the revised Supplementary Information, we have added the fitting parameters for the coordination number calculations derived from the ambient-temperature and variable-temperature PDF data. These details are provided in Supplementary Table 7 and Supplementary Table 8. Additionally, to provide an indication on the goodness of fit, we have plotted the residuals for each fitting and updated the panel (d) and (e) in Supplementary Figures 17 and Supplementary Figure 24 accordingly.

- Supplementary Table 7 added to the Supplementary Information:

Supplementary Table 7. Fitting parameters of the Gaussian fits for RT-PDF data.

Fitting parameters	(S-NEA) ₂ PbBr ₄ phase	
	crystalline	glassy

Slope of Baseline $D(r) = -4\pi\rho*r$		-1.0942	
	Peak value	2.99	2.95
Gaussian fit 1	Area	6.01	5.51
	FWHM	0.39	0.45
	Center	3.01	2.99
	Amplitude	14.83	11.38
	CN_{Pb-Br}	6.00	5.50
Gaussian fit 2	Area	5.10	6.19
	FWHM	0.69	0.81
	Center	4.22	4.19
	Amplitude	6.94	7.15

- Supplementary Table 8 added to the Supplementary Information:

Supplementary Table 8. Fitting parameters of the Gaussian fits for VT-PDF data.

Fitting parameters	Temperature (°C)					
	27	103	135	168	190	
Slope of Baseline $D(r) = -4\pi\rho*r$	-1.0942					
Peak value	2.99	2.98	2.98	2.97	2.93	
Gaussian Fit 1	Area	5.73	5.61	5.55	5.48	4.73
	FWHM	0.36	0.39	0.40	0.41	0.50
	Center	3.01	3.01	3.01	3.01	2.97
	Amplitude	15.00	13.68	13.19	12.60	8.81
	CN_{Pb-Br}	6.00	5.87	5.81	5.73	4.95
Gaussian Fit 2	Area	5.60	5.80	5.70	5.65	7.40
	FWHM	0.73	0.79	0.80	0.81	1.06
	Center	4.20	4.22	4.21	4.22	4.21

- Previous panel (d) and (e) in Supplementary Figure 18:

Supplementary Figure 18. Peak fitting of the first and second peak in the (d) crystalline- and (e) glassy-phase PDF (baseline extracted) using Gaussian functions.

- Revised panel (d) and (e) in Supplementary Figure 17:

Supplementary Figure 17. Peak fitting of the first and second peak in the (d) crystalline- and (e) glassy-phase PDF (baseline extracted) using Gaussian functions. For clarity, the residual lines in panels (d) and (e) were offset by -2.5 and -2, respectively. For detailed fitting parameters, see Supplementary Table 7.

- Previous Supplementary Figure 25 in the Supplementary Information:

Supplementary Figure 25. Gaussian peak fitting of variable-temperature X-ray pair distribution function data. $D(r)$ of crystalline $(S\text{-NEA})_2\text{PbBr}_4$ upon heating with data obtained at (a) 27°C, (b) 103°C, (c) 135°C, (d) 168°C and (e) 190°C. The corresponding integrals of the Pb-Br peak were calculated to be 5.73, 5.61, 5.55, 5.48 and 4.73, respectively. Based on these values, the average Pb coordination numbers were calculated to be 5.87 for 103°C, 5.81 for 135°C, 5.73 for 168°C, and 5.0 for 190°C.

- Revised Supplementary Figure 24 in the Supplementary Information:

Supplementary Figure 24. Gaussian peak fitting of variable-temperature X-ray pair distribution function data. (a) $D(r)$ of crystalline $(S-NEA)_2PbBr_4$ during heating from 27°C to 190°C. Individual $D(r)$ curve of crystalline $(S-NEA)_2PbBr_4$ upon heating at (b) 27°C, (c) 103°C, (d) 135°C, (e) 168°C and (f) 190°C, with Gaussian peak fitting. The residual lines in panels (b-f) were offset by -2.5 for clarity. For detailed fitting parameters, see Supplementary Table 8.

2) Page number 10, paragraph 3: “a marked difference observed in the glassy phase is the appearance of a second, upfield, CH₃ line in the ¹H spectrum (Fig. 2c) and correlates with neighboring ¹³C signals in the two-dimensional ¹³C{¹H} heteronuclear correlation (HETCOR) spectra (Fig. 2d-e).” Figure 2c is a ¹³C spectrum.

We thank the reviewer for pointing out this oversight. We have revised the manuscript to describe Figure 2c as the ¹³C spectrum and Supplementary Figure 21b as the ¹H spectrum.

- Previous text in manuscript on Page 10:

Nevertheless, a marked difference observed in the glassy phase is the appearance of a second, upfield, CH₃ line in the ¹H spectrum (Fig. 2c) and correlates with neighboring ¹³C signals in the two-dimensional ¹³C{¹H} heteronuclear correlation (HETCOR) spectra (Fig. 2d-e).

- Revised text in manuscript on Page 11:

Nevertheless, a notable difference observed in the glassy phase is the upfield shift of the CH₃ resonance in the ¹³C spectrum (Fig. 2c). This shift corresponds to the appearance of a second, upfield, CH₃ line in the ¹H spectrum (Supplementary Fig. 21b), which, in turn, exhibits correlations with neighboring ¹³C signals in the two-dimensional ¹³C{¹H} heteronuclear correlation (HETCOR) spectra (Fig. 2d-e).

3) There are many typographical errors in the manuscripts (eg. “terahetrz” etc.)

We appreciate the reviewer for pointing out this issue. We have thoroughly reviewed both the manuscript and the Supplementary Information and corrected all typographical errors. Specifically, “terahetrz” has been corrected to “terahertz”, along with other minor errors. For a detailed list of corrections, please refer to our response to **Reviewer #1, Comment #6**.

4) Inconsistencies in the SI figure and caption, please check the SI thoroughly. Examples: Supplementary Information 2 -optical images, Figure 11; Supplementary Information 3 — Scanning electron microscopy images with EDS analysis, Figures 12 and 13.

We thank the reviewer for bringing this issue to our attention. To ensure accuracy and consistency, we have carefully reviewed the Supplementary Information and corrected any discrepancies between the figure labels and captions. Additionally, we have updated the captions of figures and tables to accurately reflect their content. The revised Supplementary Information is provided with tracked changes for reference.

Responses to Reviewer #3:

The authors conducted a thorough investigation of the structural dynamics during the melting and glass formation processes of the 2D HOIP $(S\text{-NEA})_2\text{PbBr}_4$, which encompass short-range, some intermediate-range, and extended-range structural order. Their findings offer a deeper understanding of the structural evolution and interplay between structure and properties in HOIP glasses, thereby establishing a foundation for their future applications in advanced phase-change materials. I recommend an acceptance of this study after some revisions.

We sincerely appreciate the reviewer's positive evaluation of our study. We are pleased that the reviewer acknowledges the depth of our investigation and the broader implications of our findings for hybrid perovskite glasses. We have carefully revised the manuscript to incorporate the suggested improvements and provide further clarity where needed. Our point-by-point responses to the reviewer's comments are detailed below, and we hope that the revised version satisfactorily addresses all concerns.

1. *Ab initio* molecular dynamics (AIMD) simulations are needed to add a different view and provide useful information of the microscopic structural evolution during the transitions, along with the experimental characterizations.

We appreciate the reviewer's suggestion. We agree that AIMD simulations would offer a complementary perspective to our experimental findings, providing valuable insights into the microscopic structural evolution during glass formation process. However, as noted by the editor, incorporating AIMD simulations extends beyond the primary focus of this study, which focus on experimental characterisation. Nevertheless, we acknowledge the importance of AIMD simulations and consider them a promising avenue for future investigations to further enhance the understanding of HOIP melting and glass formation.

2. *From a microscopic viewpoint, what insights can we get regarding structural reversibility after the glass-crystal phase transition with these detailed characterizations?*

We thank the reviewer for their comment. Our X-ray total scattering, PDF analysis, and solid-state NMR studies provide key microscopic insights into the structural reversibility of $(S\text{-NEA})_2\text{PbBr}_4$ during the glass-to-crystal phase transition:

(1) Retention of short-range order: PDF analysis confirms that the local coordination environment in the glass closely resembles that of the crystalline phase, indicating that short-range order is largely preserved during melting and vitrification. Upon recrystallisation, these local motifs rearrange into the long-range periodic lattice, supporting a high degree of reversibility at the atomic scale.

(2) Recovery of structural correlations between neighbouring octahedra: The Pb...Pb correlations between neighboring octahedra remain detectable in the glass phase, though weakened compared to the crystal, suggesting partial disruption of corner-sharing connectivity. Upon recrystallisation, this connectivity is largely restored, though minor variation may persist.

(3) Role of organic cations in the transition: Solid-state NMR shows that organic (S-NEA)⁺ cations remain dynamically active in the liquid and glass states, leading to weakened hydrogen bonding and increased orientational disorder. Upon recrystallisation, the cations partially regain their ordered alignment, but their exact positioning may be influenced by kinetic trapping effects, which could contribute to variations in the final crystal structure.

(4) Dynamic Aspects of Reversibility: High-temperature NMR and terahertz spectroscopy reveal that Pb-Br framework dynamics increase significantly in the liquid state, leading to greater flexibility. In the recrystallised phase, Pb-Br dynamics revert to a more rigid configuration, but some residual local distortions may remain, potentially affecting the crystal nucleation and growth pathway.

These findings indicate that (S-NEA)₂PbBr₄ exhibits a high degree of structural reversibility upon glass-to-crystal transition, particularly in local coordination and Pb...Pb connectivity. On the other hand, subtle differences between the recrystallised and as-synthesised crystals suggest minor structural modifications. For example, DSC data (Supplementary Figure 4d) indicate that the re-melting temperature of the recrystallised sample is slightly lower than that of the as-synthesised crystal; PXRD analysis (Supplementary Figure 5 and Table 2) reveals subtle deviations in lattice parameters, suggesting potential variations in crystal packing, defect density, or microstrain. Further studies on time-dependent structural relaxation, such as annealing experiments and *in situ* PDF measurements on heating the glass, could provide additional insights into the extent of structural recovery over time, particularly in understanding whether the recrystallised structure further stabilises toward the parent crystal phase and its associated properties.

3. Fig 1b shows four phases in the second heating scan. One of them is between T_g and T_x , called viscoelastic supercooled liquid phase. Is it identical to the melting phase or can it find a unique place in Fig 4?

We thank the reviewer for their comment. Upon heating, the glassy phase of (S-NEA)₂PbBr₄ transforms into a super-cooled liquid state above T_g and before reaching T_x . Our *in situ* variable-temperature NMR experiments on heating the glass, as shown in Fig. 3c-e, indicate that the dynamics of the supercooled liquid phase differ from those of the real liquid state.

However, as we currently lack variable-temperature PDF data on heating the glass, our ability to characterise its structure in detail is limited. Consequently, we are not yet able to precisely depict the supercooled liquid phase of $(S\text{-NEA})_2\text{PbBr}_4$. To improve the clarity of Fig. 4 and emphasize the crystal-liquid-glass transition during melt-quenching, we have revised the figure caption to better reflect these considerations in the revised manuscript.

- Previous caption of Fig. 4 in manuscript on Page 16:

Fig. 4 Schematic illustration of the structural dynamics of the 2D HOIP $(S\text{-NEA})_2\text{PbBr}_4$ upon melting and glass formation. Marks (i), (ii) and (iii) denote structural correlations within an octahedron, between neighbouring octahedra, and between an octahedron and its diagonal or linear second-nearest-neighbour octahedron, respectively.

- Revised caption of Fig. 4 in manuscript on Page 17:

Fig. 4 Schematic illustration of the structural dynamics of melting and vitrification in the two-dimensional hybrid organic-inorganic perovskite $(S\text{-NEA})_2\text{PbBr}_4$. This figure depicts the transition from the crystalline to the liquid and glassy phases, highlighting key structural and dynamic changes. In the crystalline phase, PbBr_6 octahedra form a well-ordered corner-sharing network, stabilised by hydrogen bonding with $(S\text{-NEA})^+$ cations. Upon melting, long-range periodicity is lost, but short-range order persists with partial retention of Pb-Br coordination and connectivity between neighbouring octahedra. Increased molecular motion in the liquid phase leads to further structural disorder, though weak extended correlations remain. Vitrification kinetically traps the disordered liquid structure, resulting in a glass that retains key motifs of the melt. The labeled structural correlations indicate (i) within individual octahedra, (ii) between neighbouring octahedra, and (iii) between second-nearest-neighbour octahedra (either diagonal or linear).

4. Define PDF when it first appears at line 81.

We appreciate the reviewer's suggestion. To ensure clarity, we have now defined PDF (pair distribution function) at its first mention in manuscript.

- Previous text in manuscript on Page 4, line 81:

"... a combination of *in situ* variable-temperature experiments, including PDF analysis ..."

- Revised text in manuscript on Page 4:

"... a combination of *in situ* variable-temperature experiments, including pair distribution function (PDF) analysis ..."

Responses to Reviewer #4:

In this paper the authors reported in-depth study of melt-quenched glasses of a HOIP, (S-NEA)₂PbBr₄, including: (1) DSC and TGA analysis that confirms the glass transition, and show the fragility; (2) the structural characterization of the melt-quenched glass by PDF, far-IR adsorption and NMR and (3) characterization of the dynamics associated with melted and glassy state. Overall the paper provides a thorough picture of the (S-NEA)₂PbBr₄ glass and should be published with minor revision. There are several points that needs clarification so that the audience would better understand the key message of this work.

We sincerely appreciate the reviewer's positive evaluation of our work and for highlighting the key contributions of our study, including the thermal, structural, and dynamic characterisations of (S-NEA)₂PbBr₄ in its glassy and liquid phases. The reviewer's insightful feedback has been invaluable in refining our manuscript and ensuring greater clarity in conveying the key findings. Below, we provide detailed responses to each of the reviewer's comments and have made the necessary revisions to improve the manuscript accordingly.

1. The (S-NEA)₂PbBr₄ is a rather well-known material for its glass transition and as the authors have mentioned, there are several previous works on this, including one of the authors own research (ref 27). I think it would be better for the authors to discuss in more detail in the introduction section to clearly state what has been done for the (S-NEA)₂PbBr₄ glass and what information is newly provided in this paper. In this way the audience can clearly appreciate the novelty of this paper.

We sincerely thank the reviewer for their comment and suggestion. In response, we have revised the introduction section to provide a more detailed discussion of prior work on (S-NEA)₂PbBr₄ glasses, clearly highlight what has been previously established and what new insights our study contributes. We hope these revisions allow reader to better understand its novelty and significance of our work. Specifically, we have updated the following sections.

- Previous text in manuscript on Page 3:

Examples of glass-forming HOIPs include 3D [TPrA][M(dca)₃] (TPrA = tetrapropylammonium, M = Mn²⁺, Fe²⁺, Co²⁺, dca = dicyanamide)²⁵, 2D chiral (S-/R-NEA)₂PbBr₄. (NEA = 1-(1-naphthyl)ethylammonium)^{26,27}, 2D (1-MeHa)₂PbI₄ (1-MeHa = 1-methyl-hexylammonium)²⁸ and 2D (MIPA)₂PbI₄ (MIPA = N-methyl iodopropylammonium)²⁹. The novelty of liquids and glasses formed from HOIPs means that microscopic insight into the nature of their non-crystalline phases, and the fundamental atomic-scale behaviours upon melting and vitrification are extremely important. This would allow for elucidation of the underlying

structure-property relationships and would thus be of great importance for taking full advantage of their utility. Recent work by Singh *et al.*³⁰ has made progress in this area, providing insights into the local structure of (S-/R-NEA)₂PbBr₄ glasses and melts, with a particular focus on comparing metal-halide coordination and the organic-inorganic interactions between crystalline and glassy phases. Studies on the structural evolution and dynamics of 2D HOIPs during crystal-to-liquid-to-glass transition remain scarce, highlighting the necessity for further in-depth investigation.

- Revised text in manuscript on Page 4:

To date, examples of glass-forming HOIPs include 3D [TPrA][M(dca)₃] (TPrA = tetrapropylammonium, M = Mn²⁺, Fe²⁺, Co²⁺, dca = dicyanamide)²⁸, 2D chiral (S-/R-NEA)₂PbBr₄. (NEA = 1-(1-naphthyl)ethylammonium)²⁹, 2D (1-MeHa)₂PbI₄ (1-MeHa = 1-methyl-hexylammonium)³⁰ and 2D (MIPA)₂PbI₄ (MIPA = N-methyl iodopropylammonium)³¹. Among these, (S-NEA)₂PbBr₄ is one of the most extensively studied systems. Prior studies by Mitzi *et al.* have made progress in this area by investigating the reversible crystal-glass switching²⁹ and the kinetics of glass crystallisation³⁰. Our earlier work also demonstrated that both chiral (S-/R-NEA)₂PbBr₄ and its non-melting racemic analog can form glasses *via* ball-milling, offering an alternative to the conventional melt-quenching approach³². More recently, Singh *et al.* investigated the local structure of this HOIP in its molten and glassy phases, with a focus on metal-halide coordination and organic-inorganic interface interactions³³. However, these studies have primarily addressed static structural properties, leaving the structural evolution during crystal-to-liquid-to-glass transitions largely unexplored. Furthermore, research into the dynamics of melting and vitrification remains scarce, yet understanding the atomic-scale behaviours during these processes is essential. These insights not only enable the elucidation of the underlying structure-property relationships but also pave the way for unlocking their full potential in advanced applications such as phase-change memory and computing²⁵.

- Previous text in manuscript on Page 4:

In this article, we focus on the melting and glass formation of a prototypical glass-forming 2D HOIP, (S-NEA)₂PbBr₄. The microstructure of (S-NEA)₂PbBr₄ in both its crystalline and glassy phases was investigated using scanning transmission electron microscopy (STEM). We then compared their atomic structure and reveal the degree of structural disorder across various length scales in the glassy phase through X-ray total scattering and multidimensional magic angle spinning (MAS) solid-state nuclear magnetic resonance (ssNMR). Furthermore, a combination of *in situ* variable-temperature experiments, including PDF analysis, terahertz far-infrared (THz/Far-IR) absorption spectroscopy and multinuclear ssNMR, was employed to yield

a complete picture of the melting mechanism, and provide new insights into the structure and dynamics of the HOIP melts. Finally, we extend our discussion to the optical and mechanical properties of the glassy material, establishing new structure-property relationships.

- Revised text in manuscript on Page 4:

In this article, we build upon previous studies to present a comprehensive investigation into the structural dynamics of $(S\text{-NEA})_2\text{PbBr}_4$ during melting and glass formation. The microstructure of both the crystalline and glassy phases was examined using scanning transmission electron microscopy (STEM). The degree of structural disorder across various length scales in the glassy phase was further elucidated through X-ray total scattering and multidimensional magic angle spinning (MAS) solid-state nuclear magnetic resonance (ssNMR). To yield a complete picture of the melting mechanism, we employed a combination of *in situ* variable-temperature experiments, including pair distribution function (PDF) analysis, terahertz far-infrared (THz/Far-IR) absorption spectroscopy, and multinuclear ssNMR. These complementary techniques provided, for the first time, detailed insights into the structural evolution and transition dynamics during the crystal-to-liquid-to-glass transformation. Finally, we extend our investigation to correlate the optical and mechanical properties of the glassy material with its structural features, thereby establishing new structure-property relationships. By integrating these analyses, this work sheds light on the microscopic mechanisms underpinning the glass formation in HOIPs and paves the way for the rational design of hybrid glasses with advanced functional properties.

2. In ref 27, the authors reported that the $(S\text{-NEA})_2\text{PbBr}_4$ can also give a glass by ball-milling, which has lower T_g . Could the author comments on the coordination number of the ball-milled glass compared to the melt-quenched glass? similarly, it would be good to show the comparison in short and medium range ordering.

We thank the reviewer for their comment and interest in our previous work on the $(S\text{-NEA})_2\text{PbBr}_4$ glass formed *via* ball-milling (*Chem. Sci.*, 2024, 15, 7198–7205). To address this point, we have added Supplementary Figure 19 to the Supplementary Information, which compares the total scattering data of glasses formed by melt-quenching and ball-milling.

As this study primarily focuses on the structural dynamics of the melting and quenching processes, an in-depth discussion on ball-milled glasses would extend beyond the main scope of this work. Nevertheless, we acknowledge the importance of this comparison and have included a brief discussion in the manuscript. We also agree that understanding how different glass formation methods influence structural ordering is a valuable topic that warrants further

investigation in future studies.

- Supplementary Figure 19 added to the Supplementary Information:

Supplementary Figure 19. Total scattering data for ball-milled glass $a_m(\text{S-NEA})_2\text{PbBr}_4$, obtained from our previous study² and processed using the same condition as those for the melt-quenched glass $a_g(\text{S-NEA})_2\text{PbBr}_4$. (a) Comparison of the structure factor $S(Q)$ between $a_g(\text{S-NEA})_2\text{PbBr}_4$ and $a_m(\text{S-NEA})_2\text{PbBr}_4$, with the inset showing the corresponding low- Q feature. Comparison of the pair distribution functions $D(r)$ between $a_g(\text{S-NEA})_2\text{PbBr}_4$ and $a_m(\text{S-NEA})_2\text{PbBr}_4$, in the range of (b) 0-15 Å and (c) 7-14 Å, highlighting their similarity in the short-range and intermediate-range ordering. (d) Gaussian peak fitting of the pair distribution function for $a_m(\text{S-NEA})_2\text{PbBr}_4$. The residual line was offset by -2 for clarity. The integral of the Pb-Br peak were calculated to be 5.62, corresponding to an average Pb coordination number of *ca.* 5.6 for the ball-milled glass. This is comparable to that of the melt-quenched glass, which has a value of *ca.* 5.5.

- Section added to manuscript in Page 4:

Our earlier work also demonstrated that both chiral (*S*/*R*-NEA)₂PbBr₄ and its non-melting racemic analog can form glasses *via* ball-milling, offering an alternative to the conventional

melt-quenching approach³².

- Section added to manuscript in Page 11:

In addition, to further explore how different preparation routes influence structural ordering, we compared the total scattering data of glasses formed *via* melt-quenching and ball-milling. The PDF data of the ball-milled glass, $a_m(S\text{-NEA})_2\text{PbBr}_4$ (a_m = mechanically amorphised), obtained from our prior study³², reveals similar locally coordinated structural motifs to those in $a_g(S\text{-NEA})_2\text{PbBr}_4$, with comparable Pb-Br coordination numbers (Supplementary Fig. 19). While these findings offer initial insights, further exploration of how different glass formation methods affect the overall structure and physical properties of HOIP glasses remains an intriguing avenue for future research.

3. The quenching speed should also affect the medium range ordering of the Pd octahedral. Can the authors compare glasses made with different quenching rate?

We appreciate the reviewer's comment, and we acknowledge that investigating the effects of varying quenching rates on the local structure of glass is an important and valuable area of research. However, our current study primarily focuses on characterising glasses formed under a single quenching condition, and we do not have the relevant pair distribution data for glasses at varying quenching rates. Furthermore, conducting a systematic investigation into the effects of quenching rate would require precise thermal control and specialised setups capable of rapid quenching the capillary in the total scattering experiment. This remains a technical challenge and is beyond the scope of the current study.

To provide clarity on our experimental approach, and to support future work that may explore the influence of quenching rates, we have added a section in the manuscript detailing the preparation method used in our characterisations. We hope this addition will be useful for researchers interested in this topic.

- Section added to manuscript on Page 21:

Preparation of (S-/R-NEA)₂PbBr₄ glasses: Unless otherwise stated, all glassy samples characterised in the study were prepared using the same method adapted from the literature²⁹. The dried crystals were ground into fine powders using a mortar and pestle, and placed on a clean glass substrate. The substrate, along with the powders, was placed on a preheated hot plate at 190 °C for approximately two minutes, until the solid powders were visually observed to have fully transformed into the liquid state. The glass substrate was then removed from the hot plate using tweezers, quickly covered with another room-temperature clean glass slide, and placed on a room-temperature metallic bench. Once fully cooled, the

glassy sample was scraped from the glass substrate surface using a spatula. It should be noted that the above experimental procedures need to be carried out in a fume hood.

4. In the mediate range regime, it seems that the 2D feature of the HOIP is preserved to a certain extent in the glassy state. If the same is melt-quenched under pressure (hot-press), would the resulting glass show anisotropic behavior?

We appreciate the reviewer's insightful question. All glassy samples in our study were prepared *via* melt-quenching without applied external pressure. The material was melted on a hot plate and rapidly cooled on a room-temperature metallic bench. Our structural analysis indicates that while long-range periodicity of $a_g(S\text{-NEA})_2\text{PbBr}_4$ is lost upon melt-quenching, some degree of residual layered arrangement persists in the glassy phase, as evidenced by X-ray total scattering and pair distribution function analysis.

Given this residual structural correlation, applying external pressure during melt-quenching could potentially enhance the alignment of these motifs, leading to anisotropic behavior in the glass. If pressure promotes preferential orientation of the remaining layered structures, it could influence direction-dependent optical, mechanical, or vibrational properties. Investigating this possibility through polarised Raman spectroscopy and birefringence measurements would be valuable in determining whether directional structuring can be induced in the glass. This remains an intriguing avenue for future research, offering potential insights into how processing conditions can be used to tailor the properties of hybrid perovskite glasses.

5. The authors reported several characterizations, but the results are associated with several structural features (Pd octahedral distortion, etc). Could the author add one paragraph that discuss their findings in the order of structure features (i.e. firstly on Pd octahedral, then on Pd..Pd, then on organic portion, finally on the dynamics)? This will make the picture more clear.

We appreciate the reviewer's insightful suggestion, and we have revised the manuscript to provide a clearer discussion of the structural and dynamic evolution of $(S\text{-NEA})_2\text{PbBr}_4$ upon melting and glass formation. Below, we summarise the key structural features in a systematic manner, progressing from individual PbBr_6 octahedra, to $\text{Pb}\dots\text{Pb}$ connectivity between neighbouring octahedra, the role of organic cations, and finally, the dynamic behaviors.

- Section added to manuscript on Page 16:

To establish a comprehensive understanding of the melting and glass formation process, Fig. 4 provides a visual summary of the key structural and dynamic changes in $(S\text{-NEA})_2\text{PbBr}_4$. In the crystalline phase, it adopts a layered perovskite structure, where PbBr_6 octahedra form an

extended corner-sharing network with sixfold coordination. Upon melting, long-range periodicity is lost, while short-range order persists. The Pb-Br coordination number decreases to an average of approximately 5.0, indicating slight decoordination in the melt, while Pb...Pb correlations between neighbouring octahedra remain detectable despite weakening, suggesting partial retention of corner-sharing connectivity. Structural disorder increases beyond three neighbouring octahedra, though extended correlations up to 80 Å in the PDF suggest the presence of organic-metal-organic units or remnants of layered order. Upon quenching, the glass retains the structural motifs of the melts, with a slight increase in Pb-Br coordination, reflecting a coexistence of five- and six-coordinated Pb environments. The (S-NEA)⁺ cations play a crucial role in stabilising the perovskite framework through hydrogen bonding. Upon melting, these interactions weaken as the organic cations gain mobility. Despite this, the distribution of cations remains confined within the framework, preventing complete structural collapse. In the glass, the cations are kinetically trapped in a disordered yet non-random configuration, with some interactions with the inorganic layers retained. The dynamic behaviour of both inorganic and organic components evolves significantly. In the crystal, the motion of both the (S-NEA)⁺ cations and the Pb-Br framework is relatively constrained. The Pb-Br stretching modes remain rigid at low temperatures but undergo a pronounced redshift approaching the melting point, indicating increased flexibility within the inorganic framework. In the liquid, high-temperature NMR reveals sharp ¹H and ¹³C signals, suggesting rapid cation reorientation, while ²⁰⁷Pb NMR shifts and peak narrowing confirm increased Pb mobility, consistent with THz/Far-IR and PDF analysis. Upon quenching, dynamic motion is arrested, though NMR reveals residual cation mobility, particularly in the methyl and ammonium groups, whereas the Pb-Br framework exhibits restricted atomic rearrangement, with partial retention of corner-sharing connectivity.

The progressive loss of order, from the highly structured crystalline phase to the more fluid-like liquid state and ultimately to the kinetically trapped glass, underscores the complex nature of phase transitions in hybrid perovskites. Understanding these mechanisms not only advances our fundamental knowledge of hybrid perovskite glasses but also provides insight into tuning their properties for potential applications.

Additionally, we have revised the caption of Figure. 4 to better reflect the context.

- Previous caption of Figure 4 in manuscript on Page 16:

Fig. 4 Schematic illustration of the structural dynamics of the 2D HOIP (S-NEA)₂PbBr₄ upon melting and glass formation. Marks (i), (ii) and (iii) denote structural correlations within an

octahedron, between neighbouring octahedra, and between an octahedron and its diagonal or linear second-nearest-neighbour octahedron, respectively.

- Revised caption of Figure 4 in manuscript on Page 17:

Fig. 4 Schematic illustration of the structural dynamics of melting and vitrification in the two-dimensional hybrid organic-inorganic perovskite $(S\text{-NEA})_2\text{PbBr}_4$. This figure depicts the transition from the crystalline to the liquid and glassy phases, highlighting key structural and dynamic changes. In the crystalline phase, PbBr_6 octahedra form a well-ordered corner-sharing network, stabilised by hydrogen bonding with $(S\text{-NEA})^+$ cations. Upon melting, long-range periodicity is lost, but short-range order persists with partial retention of Pb-Br coordination and connectivity between neighbouring octahedra. Increased molecular motion in the liquid phase leads to further structural disorder, though weak extended correlations remain. Vitrification kinetically traps the disordered liquid structure, resulting in a glass that retains key motifs of the melt. The labeled structural correlations indicate (i) within individual octahedra, (ii) between neighbouring octahedra, and (iii) between second-nearest-neighbour octahedra (either diagonal or linear).

Responses to Reviewer #5:

This is a thorough and interesting paper on the structural changes during melting and glass formation of a two-dimensional hybrid perovskite (S-NEA)₂PbBr₄. A wide range of analytical techniques are used for compositional and structural analyses, as well as selected thermal and mechanical characterization. The work provides detailed insights into the structural dynamics of the studied materials. While the synthesized material may not have any immediate applications and other hybrid perovskite glasses are already known, I think the fundamental understanding gained on the liquid and glass states presented here makes the work suitable for publication in Nature Communications after some revisions. In detail, I hope the authors can address the following points to strengthen their manuscript:

We sincerely appreciate the reviewer's positive evaluation of our work and their recognition of the comprehensive approach we have taken to investigate the structural dynamics of (S-NEA)₂PbBr₄ during melting and glass formation. We are grateful for the constructive comments that have guided us in strengthening our manuscript. Below, we provide detailed responses to address each of the reviewer's points, and we hope that our revisions effectively enhance the clarity and impact of our study.

1. Some more discussion and clarification of the structural order of the studied liquid and glasses at different length scales is needed. In the Abstract, the authors refer to the materials having "limited short-range, some intermediate-range, and extended-range structural order". Glasses typically feature short-range order and by definition, no long-range (extended-range) order.

We thank the reviewer for their thoughtful suggestions. To improve clarity and avoid ambiguity, we have revised the relevant statement in manuscript. Instead of referring to "short-range, intermediate-range, and extended-range order," we can explicitly describe the structural correlations at different length scales, linking them to specific structural motifs, which are shown as follows.

- Previous text in the abstract on Page 2:

Our structural studies reveal that the liquid and glass formed from the 2D HOIP exhibit network-forming behaviors, featuring limited short-range, some intermediate-range, and extended-range structural order.

- Revised text in the abstract on Page 2:

Our structural studies reveal that the liquid and glass formed from the 2D HOIP exhibit network-forming behaviours, featuring limited short-range order within individual octahedra,

partial retention of metal-halide-metal connectivity between neighbouring octahedra, and residual structural correlations mediated by organic cations.

2. The novelty of the findings and the motivation for the chosen experimental design could be better highlighted at the end of the Introduction section.

We appreciate the reviewer's suggestion. To address this, we have revised the Introduction section to better emphasise the novelty of our findings and the motivations behind our experimental design. Further details on these revisions can be found in our response to **Reviewer #4, Comment #1**.

3. Figure 1b shows the melting temperature of the original crystal in "upscan 1" to be at 176 C, while that of the crystal that forms upon re-heating the glass in "upscan 2" is at 115 C. Why is this melting temperature different? As I understand from the text, the same crystal structure as the original one forms upon crystallization of the glass. In any case, recrystallization of hybrid glasses remains fairly unusual, so I am wondering if the authors have tried to prepare glass-ceramics from these materials? (although I understand it may be out of scope for this work).

We thank the reviewer for their comment and suggestion. Upon re-heating the glass (see upscan 2 in Fig.1b), recrystallisation occurs at 115°C, followed by re-melting at approximately 170°C. The melting temperature of the recrystallised material (*ca.* 170 °C) is slightly lower than that of the parent crystalline material (*ca.* 176 °C). Similar phenomena can be also observed in other hybrid glass-forming systems such as the 3D HOIP [TPrA][Mn(dca)₃] (*Nat. Chem.*, 2021, 13, 778-785). The lower melting temperature of the recrystallised sample, compared to the as-synthesised crystal, may be attributed to several factors:

(1) Thermal contact: During the upscan 1, the as-synthesised crystalline sample melts and spreads, creating a greater contact area with the DSC crucible. This improved thermal contact between the glassy sample and the crucible in subsequent upscan 2 may partially account for the shift in the measured melting temperature.

(2) Crystallite size effects: Recrystallised materials often consist of smaller crystallites, as nucleation and growth occur more rapidly compared to conventional synthesis. The higher surface-to-volume ratio of these smaller crystallites increases surface energy, which can lead to a lower temperature.

(3) Metastable polymorphism: Recrystallisation from the glassy phase may lead to the formation of a metastable polymorph with a slightly different packing arrangement, which could contribute to a lower melting temperature compared to the parent crystalline state.

Regarding the reviewer’s question about glass-ceramics, it extends beyond the scope of the present work. Nevertheless, we acknowledge that glass-ceramic formation in hybrid perovskite glasses is an intriguing area of research, particularly given the recrystallisation behaviour observed in this system. We agree that this represents a promising direction for future investigations.

4. In Supplementary Table 5, what is the origin for the large difference in the elemental concentrations of “theoretical” vs. the measured ones, although the Br/Pb molar ratios are fine? Also, what is the meaning of the * symbol next to C and N in the table?

We appreciate the reviewer’s comment and have addressed their concerns as follows:

(1) Difference in elemental concentrations: The observed differences between the “theoretical” and measured elemental concentrations for certain elements arise primarily from experimental limitations of the EDS technique. EDS has a lower sensitivity for detecting lighter elements such as carbon (C) and nitrogen (N), leading to an underestimation of their concentrations compared to the theoretical values. Despite this, the Br/Pb molar ratios align well with the expected stoichiometry, confirming that the measurements for the heavier elements (Br, Pb) are reliable.

(2) Meaning of * Symbol: The * symbol next to C and N in Supplementary Table 5 indicates that the measured concentrations for these elements are less reliable due to the limitations of the EDS technique mentioned above. This is now clarified in the revised table caption.

- Revised Supplementary Table 5 in the Supplementary Information:

Supplementary Table 5. Elemental analysis of $(S-NEA)_2PbBr_4$ and $a_g(S-NEA)_2PbBr_4$ calculated from SEM-EDS data.

Substance	C* (wt %)	N* (wt %)	Br (wt %)	Pb (wt %)	Br : Pb (molar ratio)
Theoretical	33.05	3.21	36.68	23.78	4.0
$(S-NEA)_2PbBr_4$	78.22	2.19	11.66	7.93	3.81
$a_g(S-NEA)_2PbBr_4$	80.98	2.48	9.71	6.83	3.69

* The concentrations of C and N are underestimated due to the lower sensitivity of EDS for lighter elements.

5. The differential correlation functions in Figure 2b shows that the intensity of peak II (Br-Br) is same in crystal and glass, but those of peaks I (Pb-Br) and III (Pb-Pb) are smaller in the glass.

Do the authors have an explanation for this?

We thank the reviewer for their insightful comment. As shown in Figure 2b, significant peak broadening is observed in the glassy phase, due to increased structural disorder. This broadening affects the resolution of individual correlations, particularly for overlapping peaks.

Peak II, primarily attributed to the Br-Br correlation, is partially overlapped with peaks I and III, and this overlap becomes more pronounced in the glass due to the broadening. The apparent intensity of peak II remains similar between the crystalline and glassy phases, but this does not necessarily imply that the Br-Br correlations are unaffected. Instead, it is likely that broadening and peak overlap obscure subtle changes, making them difficult to resolve without accurate quantitative deconvolution. Additionally, interactions involving the organic (*S*-NEA)⁺ cations may contribute to the observed peak intensities, further complicating the direct intensity comparison between different phases. Although challenging, a more detailed analysis to disentangle these contributions would be valuable, and we acknowledge that this is an interesting direction for future research.

6. To improve clarity for readers (at least for me), I would have preferred to have the PDF data collected at the P02.1 beamline (p. 11) presented together with the other PDF data collected at the I15-1 beamline (pp. 7-8). It would be interesting to understand the liquid-PDF data before glass-PDF data. As a minor note, it is also not clear why high-temperature total scattering data had to be collected at two different beamlines.

We thank the reviewer for their comment and suggestion. To enhance clarity, we have added Supplementary Figure 29 to compare the total scattering data collected from two different beamlines, including the I15-1 beamline at the Diamond Light Source ($\lambda = 0.161669 \text{ \AA}$) and the P02.1 beamline at PETRA III, DESY ($\lambda = 0.20734 \text{ \AA}$).

Using the I15-1 beamline, we performed the room-temperature (RT) total scattering experiments to collect data from the as-synthesised crystalline sample and the melt-quenched glass sample. At the same time, we conducted high-temperature (HT) experiments on the crystalline sample at 180°C and 190°C, both of which exceed its melting temperature, to investigate the structure of the liquid phase at different temperatures. To further understand the structural evolution of (*S*-NEA)₂PbBr₄ upon heating, we performed variable-temperature (VT) total scattering experiments on the crystalline sample over a temperature range of 27°C to 190°C. These measurements were carried out using the P02.1 beamline. The decision to use two beamlines was primarily due to beamtime limitations and the temporary shutdown of the I15-1 beamline during our scheduled experiments.

- Supplementary Figure 29 added to the Supplementary Information:

Supplementary Figure 29. Comparison of X-ray total scattering data of $(S-NEA)_2PbBr_4$ collected from different beamlines*: (a) Structure factor $S(Q)$; (b) Pair distribution function $D(r)$. Room-temperature (RT) total scattering experiments were conducted using the I15-1 beamline at the Diamond Light Source (Oxfordshire, UK) with a wavelength of $\lambda = 0.161669 \text{ \AA}$, on both the as-synthesised crystalline sample and the melt-quenched glass sample. High-temperature (HT) total scattering experiments were also performed on the crystalline sample at 180°C and 190°C - both above its melting temperature - to probe the liquid-state structure. To investigate the structural evolution of $(S-NEA)_2PbBr_4$ upon heating, variable-temperature (VT) total scattering experiments were carried out from 27°C to 190°C using the PETRA III P02.1 beamline at DESY (Hamburg, Germany) with a wavelength of $\lambda = 0.20734 \text{ \AA}$.

*The use of two beamlines was necessitated by beamtime availability.

7. The statement “temperature only has a small effect on the local ordering of melts” on p. 12 is not clear to me, when a 17% reduction in Pb-Br coordination number upon melting is reported on p. 11 based on the present data.

We thank the reviewer for their comment. To address this point and ensure clarity, we have rephrased the statement to better align the observations and the context of the data.

- Previous text in manuscript on Page 12:

In the short- and intermediate-*r* regions, the liquid-state PDFs at different temperatures are mostly identical (Supplementary Fig. 28a-b), suggesting that temperature only has a small effect on the local ordering of melts.

- Revised text in manuscript on Page 13:

In the short- and intermediate-*r* regions, the PDFs of the liquid phases show minimal changes across different temperatures (Supplementary Fig. 29a-b), indicating that further heating has little effect on the local structural ordering within the melt.

8. Why does the ^{207}Pb MAS NMR signal appear to vanish at high temperature (Supplementary Figure 32), while it is stated in the text on p. 14 when discussing these data that “the signal in the melt appears much more pronounced and substantially sharper relative to the solid”? In comparison, the intensity of the ^{12}C MAS NMR peak remains in the liquid state (Supplementary Figure S31). Also, the inset shown at top of Supplementary Figure 32 is not explained in the figure caption.

We thank the reviewer for their comment. The magnetization (and thus the signal intensity) of ^{207}Pb nuclei follows the Curie law, decreasing with increasing temperature. As the crystalline material transitions into the liquid state, the reduced magnetization results in an apparent loss of signal intensity in the ^{207}Pb MAS NMR spectra (previous Supplementary Figure 32, current Supplementary Figure 33). These signals are not normalised, making the decrease in signal intensity more pronounced. In contrast, ^1H and ^{13}C MAS NMR spectra (previous Supplementary Figures 30 and 31, current Supplementary Figures 31 and 32) are normalised, so the same Curie-law behaviour manifests as a decrease in signal-to-noise ratio rather than a visible loss of the signal.

While all nuclei experience this temperature-dependent reduction in magnetization, the non-normalised ^{207}Pb data visually exaggerate the effect. To reflect this, we have revised the captions of Supplementary Figures 30-33 in Supplementary Information. Regarding the inset in Supplementary Figure 32, it displays an enlarged view of the ^{207}Pb signal between 700 and

800 ppm, highlighting the enhanced mobility of Pb nuclei in the melt at 185°C. This has now been clarified in the revised figure caption.

- Previous caption of Supplementary Figure 30:

Supplementary Figure 30. *In situ* variable-temperature ^1H MAS NMR spectra for $(\text{S-NEA})_2\text{PbBr}_4$. Asterisks (*) denote spinning sidebands.

- Revised caption of Supplementary Figure 31:

Supplementary Figure 31. *In situ* variable-temperature ^1H MAS NMR spectra for $(\text{S-NEA})_2\text{PbBr}_4$, recorded upon heating from 27°C to 185°C and subsequent cooling to the quenched glass. All ^1H NMR signals are normalised and asterisks (*) denote spinning sidebands.

- Previous caption of Supplementary Figure 31:

Supplementary Figure 31. *In situ* variable-temperature ^{12}C MAS NMR spectra for $(\text{S-NEA})_2\text{PbBr}_4$.

- Revised caption of Supplementary Figure 32:

Supplementary Figure 32. *In situ* variable-temperature ^{13}C MAS NMR spectra for $(\text{S-NEA})_2\text{PbBr}_4$, recorded upon heating from 27°C to 185°C and subsequent cooling to the quenched glass. All ^{13}C NMR signals are normalised.

- Previous caption of Supplementary Figure 32:

Supplementary Figure 32. *In situ* variable-temperature ^{207}Pb MAS NMR spectra for $(\text{S-NEA})_2\text{PbBr}_4$ upon heating.

- Revised caption of Supplementary Figure 33:

Supplementary Figure 33. *In situ* variable-temperature ^{207}Pb MAS NMR spectra for $(\text{S-NEA})_2\text{PbBr}_4$, recorded upon heating from 27°C to 185°C. The inset at the top highlights the narrowing of the ^{207}Pb signal at 185°C, suggesting enhanced mobility of Pb nuclei in the melt.

9. How sensitive are the reported mechanical properties to the assumption of a Poisson's ratio of 0.2? I would recommend reporting the reduced modulus instead of the Young's modulus to avoid relying on an assumed value without further evidence.

We thank the reviewer for their suggestion. In our study, a Poisson's ratio of $\nu = 0.2$ was used in accordance with previous literature on hybrid organic-inorganic perovskites (*Nat. Chem.*,

2021, 13, 778-785 and *ACS Appl. Mater. Interfaces*, 2021, 13, 31642-31649). We agree that reporting the reduced modulus (E_r) is essential in the absence of a directly measured Poisson's ratio. To address this, we have revised the manuscript to include both E_r and Young's modulus (E) values, along with a discussion on the sensitivity of E to the assumed Poisson's ratio.

Specifically: (1) The revised manuscript now presents E_r as the primary reported mechanical property, with E calculated separately, using $\nu = 0.2$ for reference. (2) A sensitivity analysis has been added, showing that increasing ν from 0.2 to 0.3 (a 50% increase) results in only a ~5% change in E , indicating that the Young's modulus is not highly sensitive to the assumed Poisson's ratio. (3) Supplementary Table 10 now compares E_r , E , and hardness (H) for both the crystalline and glassy phases of $(S\text{-NEA})_2\text{PbBr}_4$.

- Previous text in manuscript on Page 15:

... the Young's moduli (E) and harness (H) properties of the crystalline phase were measured with the (0 0 1) lattice plane facing up. Average values of E and H were recorded from the load-displacement data of indentations between 1000 nm and 2000 nm, showing $E = 2.3$ GPa, $H = 0.12$ GPa for $(S\text{-NEA})_2\text{PbBr}_4$ and $E = 5.3$ GPa, $H = 0.25$ GPa for $\alpha_g(S\text{-NEA})_2\text{PbBr}_4$ (Supplementary Fig. 38).

- Revised text in manuscript on Page 18:

... the reduced modulus (E_r) and hardness (H) were measured from the load-displacement data of indentations with the (0 0 1) lattice plane facing up. The E_r and H values of $\alpha_g(S\text{-NEA})_2\text{PbBr}_4$ were significantly higher than those of its crystalline counterparts (Supplementary Table 10). To estimate the Young's modulus (E), a Poisson's ratio of 0.2 was assumed, resulting in $E = 2.30 \pm 0.10$ GPa for $(S\text{-NEA})_2\text{PbBr}_4$ and $E = 5.29 \pm 0.04$ GPa for $\alpha_g(S\text{-NEA})_2\text{PbBr}_4$.

- Section added to manuscript on Page 25:

Based on the Oliver-Pharr approach, the reduced modulus (E_r) was calculated from the indentation curves (Supplementary Table 10). To determine the Young's modulus (E) of the sample, we used the E of the diamond indenter as 1141 GPa and its Poisson's ratio (ν) as 0.07, while assuming $\nu = 0.2$ for the sample, following previous studies on hybrid perovskites^{28,65}. We also evaluated the sensitivity of E to variations in ν , finding that increasing ν to 0.3 (as reported for lead iodide HOIPs⁶⁶) resulted in only a 5% difference in the corresponding E values. This confirms that the mechanical property estimates remain robust to reasonable variations in Poisson's ratio assumptions.

- Supplementary Table 10 added to the Supplementary Information:

Supplementary Table 10

Comparison of the mechanical properties for $(S\text{-NEA})_2\text{PbBr}_4$ crystal and glass.

Sample	Reduced modulus E_r (GPa)	Young's modulus E (GPa) ^b	Hardness (GPa)
Crystal ^a	2.39 ± 0.10	2.30 ± 0.10	0.12 ± 0.01
Glass	5.48 ± 0.04	5.29 ± 0.04	0.25 ± 0.01

^a Measured with the (0 0 1) lattice plane facing up.

^b Assumes Poisson's ratio $\nu = 0.2$. Sensitivity analysis shows that increasing ν to 0.3 (a 50% increase) results in only a $\sim 5\%$ change in E .

10. Figure 4 is a nice schematic presentation of the structural insights obtained on the various phases. However, the figure is not explained or discussed in much detail in the text. I think this would be important to do to assist readers in making the connections between the results of the different characterization methods.

We appreciate the reviewer's suggestion. To enhance clarity and help readers make better connections between the results from different characterisation methods, we have added one paragraph in the main text to provide a more detailed discussion Figure 4. Further details on these revisions can be found in our response to **Reviewer #4, Comment #5**. We hope these updates enhance the understanding of Figure 4 in the context of the study.

Additional minor points:

11. Abbreviation PDF should be defined on p. 4.

We appreciate the reviewer's comment, and we have revised the manuscript to define PDF (pair distribution function) at its first mention to ensure clarity.

- Previous text in manuscript on Page 4:

... a combination of *in situ* variable-temperature experiments, including PDF analysis ...

- Revised text in manuscript on Page 4:

... a combination of *in situ* variable-temperature experiments, including pair distribution function (PDF) analysis ...

12. "terahetrz" -> "terahertz" on p. 4

We thank the reviewer for pointing out this issue. We have carefully reviewed the manuscript and the Supplementary Information and corrected all typographical errors, including “terahetrz,” which has been revised to “terahertz.” For further details, please refer to our response to **Reviewer #1, Comment #6**.

13. Supplementary Figures 23a, 25a, 26a: colors of the different lines (temperatures) are hard to distinguish.

We thank the reviewer for pointing out this issue. To ensure better visual contrast, we have updated the colour scheme in Supplementary Figures.

14. Authors refer to a Supplementary Figure 39 on p. 16, but I could not find it. They seem to be referring to Supplementary Figure 38e?

We thank the reviewer for pointing out this discrepancy. The reference to Supplementary Figure 39 on page 16 was incorrect and should refer to Supplementary Figure 38e. We have referred to the correct figure in the revised manuscript.

RESPONSE TO REVIEWERS' COMMENTS

Responses to Reviewer #1:

My previous comments and suggestions were appropriately taken into account by the authors. Many thanks for this very nice piece of work!

We sincerely thank the reviewer for the kind words and positive feedback. We are pleased that the revised manuscript has addressed all previous concerns.

Responses to Reviewer #2:

Authors have revised the manuscript to my satisfaction. The additional results and critical structure property correlation has significantly improved the scientific rigour of the present work. I recommend the publication of the revised manuscript.

We greatly appreciate the reviewer's positive feedback. We are grateful for the recommendation to accept the manuscript in its current form.

Responses to Reviewer #3:

The manuscript has been carefully revised. I think it can be accepted in the present form.

We sincerely thank the reviewer for the positive feedback and the recommendation to publish the manuscript.

Responses to Reviewer #4:

The authors have addressed my concerns and I support the publication of this paper.

We greatly appreciate the reviewer's positive feedback and support for publication.

Responses to Reviewer #5:

The authors have addressed all the comments satisfactorily. I recommend acceptance of this version for publication.

We sincerely thank the reviewer for the positive feedback and the recommendation to accept the revised manuscript.